# MIRAGE: EVALUATING AND EXPLAINING INDUCTIVE REASONING PROCESS IN LANGUAGE MODELS

**Jiachun Li[1,2], Pengfei Cao[1,2], Zhuoran Jin[1,2], Yubo Chen[1,2,*], Kang Liu[1,2], Jun Zhao[1,2]**
[1]School of Artificial Intelligence, University of Chinese Academy of Sciences
[2]The Key Laboratory of Cognition and Decision Intelligence for Complex Systems,
Institute of Automation, Chinese Academy of Sciences
{jiachun.li, pengfei.cao, zhuoran.jin, yubo.chen}@nlpr.ia.ac.cn

## ABSTRACT

Inductive reasoning is an essential capability for large language models (LLMs) to achieve higher intelligence, which requires the model to generalize rules from observed facts and then apply them to unseen examples. We present MIRAGE, a synthetic dataset that addresses the limitations of previous work, specifically the lack of comprehensive evaluation and flexible test data. In it, we evaluate LLMs' capabilities in both the inductive and deductive stages, allowing for flexible variation in input distribution, task scenario, and task difficulty to analyze the factors influencing LLMs' inductive reasoning. Based on these multi-faceted evaluations, we demonstrate that the LLM is a poor rule-based reasoner. In many cases, when conducting inductive reasoning, they do not rely on a correct rule to answer the unseen case. From the perspectives of different prompting methods, observation numbers, and task forms, models tend to conduct correct deduction without correct inductive rules consistently. Besides, we find that LLMs are good neighbor-based reasoners. In the inductive reasoning process, the model tends to focus on observed facts that are close to the current test example in feature space. By leveraging these similar examples, the model maintains strong inductive capabilities within a localized region, significantly improving its reasoning performance.

## 1 INTRODUCTION

Inductive reasoning, known as the ability of an intelligent agent to infer abstract rules from limited observations and apply them to new examples, is crucial for large language model (LLMs) (Li et al., 2023; Wen et al., 2023; Sun et al., 2024a) progressing toward artificial general intelligence (AGI) (Xu et al., 2024b; Sun et al., 2024b; Wang et al., 2024b). As illustrated in Figure 1, given a set of observed facts, inductive reasoning process expect the model to generate abstract rules from the provided facts (i.e. [A,B,C] $\rightarrow$ [B+C,B+C,C] in the rule induction task) and apply these rules to answer specific new questions (i.e. [3,4,7] $\rightarrow$ [11,11,7] in the example inference task). Despite its significant research value, it has been relatively neglected compared to other types of reasoning (e.g., math reasoning, multi-hop reasoning, etc.).

Recently, some works have started to explore this problem. They primarily evaluate the model's inductive reasoning capabilities using various datasets (Shao et al., 2024; Cheng et al., 2024; Qiu et al., 2024; Jiang et al., 2024). Though they have made great progress, their works still have two main limitations: **(1) Previous works lack comprehensive evaluation.** Most works have only one evaluation task: the inductive task on collected rules (Yang et al., 2024b; Shao et al., 2024; Wang et al., 2025) or the deductive task on specific test samples (Chollet, 2019; Xu et al., 2024a; Qiu et al., 2024). Therefore, they can only evaluate the rule induction performance or final results of inductive reasoning, instead of comprehensively analyzing the whole process (i.e. inductive + deductive). **(2) Previous works lack flexible test data.** Most former datasets evaluate the overall performance of models by collecting observation and test examples under the same rules (Rule, 2020; Kim et al., 2022; Lake et al., 2019). However, due to the absence of transformation rules, it is impossible to extend these examples, resulting in a fixed test set. This limitation makes it challenging to assess the

---

*Corresponding authors.

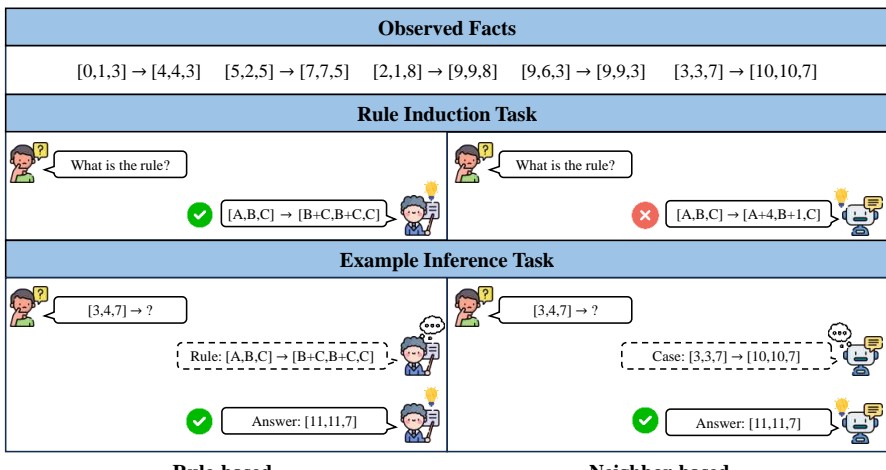

Figure 1: An overview of two paradigms (i.e. rule-based and neighbor-based) in inductive reasoning.

impact of factors such as distribution, quantity, and form of input examples on the model's inductive reasoning, thereby hindering a deeper analysis of the model's reasoning mechanisms.

In this paper, we present MIRAGE (Meta Inductive ReAsoninG Evaluation), a dataset designed to address the two aforementioned limitations. It includes both inductive and deductive evaluation tasks, while offering flexibility to construct test data with various forms, arbitrary input distributions, and controllable difficulties. In detail, we first construct a rule library based on various vector operations (e.g., [A,B,C] → [B+C,B+C,C] as shown in Figure 1). Using the automatically synthesized rules, we can generate facts arbitrarily through instantiation, ensuring the flexibility and scalability of the test data. Next, we filter out the noise data (e.g. duplicated facts) to further improve the effectiveness and quality of our dataset. Finally, to comprehensively evaluate the inductive reasoning process, we not only design inductive and deductive questions based on the synthesized data but also construct diverse application scenarios for these tasks, including list transformations, real-world problems, code generations, and string transformations (as shown in Figure 2).

Based on our dataset, we perform a deeper analysis of the model's inductive reasoning process, from which we draw two new conclusions about the inductive reasoning mechanisms of LLMs: **(1) Language models are poor rule-based reasoners.** As shown in the left column of Figure 1, in the rule-based reasoning paradigm, inductive reasoning involves first deriving the correct rule through the observation of examples and then using the inductive rule to answer new questions (like what humans do). However, we find that LLMs perform poorly in this paradigm: In many cases, though they can not induce a correct rule, they can still perform well on example inference tasks. Through experimentation, we observe this performance gap between induction and deduction across different prompting methods, models, observed example numbers, and scenarios. This indicates that the final performance of the LLM's inductive reasoning rarely relies on the intermediate inductive rules. **(2) Language models are good neighbor-based reasoners.** Furthermore, we identify an important mechanism behind LLM's inductive reasoning, which we refer to as "**neighbor-based reasoning**": If some observed facts are close to the test examples in feature space, the model tends to leverage this similarity to improve inductive reasoning performance. For example, as shown in the right column of Figure 1, even when the model cannot generate the correct rule, it can rely on the neighbor fact [3,3,7] → [10,10,7] (here the distance between [3,3,7] and [3,4,7] is small, so we refer to them as neighbors) to successfully performs the reasoning. We demonstrate that this paradigm persists across different scenarios, models, and observed example numbers. However, it can only enhance the performance within a localized scope.

To sum up, the main contributions of our work are as follows: **(1) We present a new dataset MI-RAGE**, through it, we can comprehensively evaluate the LLM's inductive reasoning process under more flexible settings. **(2) We find that LLM is a poor rule-based inductive reasoner**. In many cases, it does not rely on inductive rules to make correct deductions. **(3) We demonstrate that LLM is a neighbor-based inductive reasoner**. When performing inductive reasoning, models rely

on the neighbor facts in the observed fact set to get better performance. Our code is available at: https://github.com/BugMakerzzz/mirage.

## 2 DATA CONSTRUCTION

In this section, we describe the whole pipeline to build MIRAGE. We start by constructing rules based on five basic operations (§2.1). Next, we substitute the instantiate vectors into the rules to generate facts (§2.2) and apply filtering to them (§2.3). Finally, we transform the facts into different scenarios, creating questions to evaluate the LLM's inductive reasoning performance (§2.4).

### 2.1 RULE GENERATION

According to previous work and relevant definitions (Huber, 2017; Han et al., 2024), in inductive reasoning, for each observed fact $\mathbb{X}_k = (\boldsymbol{x}, \boldsymbol{y})$, the input vector $\boldsymbol{x}$ is transformed into the output vector $\boldsymbol{y}$ according to a certain rule $f$, i.e.:

$$f(\boldsymbol{x}) = \boldsymbol{y}, \quad \forall (\boldsymbol{x}, \boldsymbol{y}) \in \mathbb{X} \tag{1}$$

where $\mathbb{X}$ is the observed fact set under the rule $f$. Here $f$ is the core of the problem, as it allows us to generate facts for $\mathbb{X}$ based on it automatically. Conversely, inferring $f$ from $\mathbb{X}$ requires significantly more effort due to the vast range of possible rules. Therefore, we first consider automating these rules' large-scale synthesis.

Based on previous representative datasets (Chollet, 2019; Rule, 2020; Xu et al., 2024a), we summarize the main types of rules, resulting in five atomic operations in this dataset: **(1) Add:** The operation adds certain components together. For example: $[x, y, z] \rightarrow [x, x+y, z]$. **(2) Copy:** The operation copies some components to others. For example: $[x, y, z] \rightarrow [x, x, z]$. **(3) Map:** The operation applies a linear transformation to some components. For example: $[x, y, z] \rightarrow [x, ky + b, z]$. To avoid the interference of complex math calculations, we have $k \in [1, 9]$ and $b \in [0, 9]$. **(4) Pad:** The operation fills certain components with constant values. For example: $[x, y, z] \rightarrow [x, c, c]$, where $c \in [0, 9]$. **(5) Swap:** The operation swaps certain components. For example: $[x, y, z] \rightarrow [z, y, x]$.

For each operation $O$, we randomly initialize the set index vector $\boldsymbol{d}$ on which the operation applies and the index vector $\boldsymbol{r}$ where the result is output. Specifically, for $x \in \boldsymbol{x}, y \in \boldsymbol{y}$:

$$y_j = \begin{cases} [O(x_{\boldsymbol{d}})]_i, & \text{if } j \in \boldsymbol{r} \\ x_j, & \text{if } j \notin \boldsymbol{r} \end{cases} \tag{2}$$

where $r_i = j$ and $[\cdot]_i$ represents the $i$-th component. Therefore, we can generate a meta-rule $f = (O, \boldsymbol{d}, \boldsymbol{r})$. Through sampling $(O, \boldsymbol{d}, \boldsymbol{r})$ randomly, we can construct a meta-rule library $\mathbb{F}$.

### 2.2 FACT GENERATION

After generating the rule library, we can randomly initialize $\boldsymbol{x}$, and apply a specific rule $f \in \mathbb{F}$ to get $\boldsymbol{y}$. We repeat this process to generate the fact set $\mathbb{X}$ under the rule $f$. All the $(\boldsymbol{x}, \boldsymbol{y}) \in \mathbb{X}$ are used for the LLM to induce the rule $f$. It is worth noting that we can control the inductive difficulty by adjusting two factors: the dimension $D$ of $\boldsymbol{x}, \boldsymbol{y}$ and the fact number $N$ of $\mathbb{X}$. As an example, in Figure 1, $D$ is 3 and $N$ is 5. Empirically, a higher $D$ and a smaller $N$ tend to increase the task difficulty. Additionally, to avoid the interference of complex mathematical calculations in evaluating inductive reasoning ability, we restrict the elements in each $\boldsymbol{x}$ to integers between 0 and 9.[1] Since we can synthesize any $D$-dimensional vector $\boldsymbol{x}$ to construct a fact, we can flexibly control the input distribution.

### 2.3 DATA FILTERING

To ensure the quality of the dataset and the effectiveness of the evaluation, we need to filter out some noisy data. The following filtering steps are applied: **(1) Filtering out duplicate facts.** For any two facts in $\mathbb{X}$, if their input vectors $\boldsymbol{x}$ are identical, one of them is removed and resampled. This ensures that for each rule, all observed facts are unique. **(2) Filtering out duplicate rules.** To ensure diversity in the evaluation, we also remove duplicate rules, which have the same $(O, \boldsymbol{d}, \boldsymbol{r})$. **(3) Filtering out trivial facts.** After random sampling, $\mathbb{X}$ may include some trivial facts that provide little value for model induction, such as facts like $\boldsymbol{x} = \boldsymbol{y}, \boldsymbol{x} = 0$, or $\boldsymbol{y} = 0$. We filter the data to ensure that each $\mathbb{X}$ contains at most one trivial fact, thereby limiting the noise that could affect the model's inductive reasoning process.

---

[1]Our pilot experiments indicate that, under these constraints, most of the models can achieve an accuracy of nearly 100% in performing purely mathematical operations. See Appendix A.2 for details.

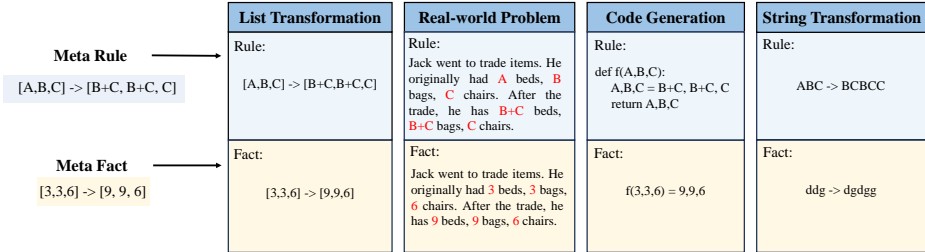

Figure 2: Examples in four different scenarios of MIRAGE.

## 2.4 QUESTION GENERATION

So far, we have constructed all the metadata that we need to generate specific questions. It is worth noting that both $\mathbb{F}$ and $\mathbb{X}$ contain only abstract rules and facts, without any specific context. Therefore, they represent the fundamental inductive reasoning test data, which is why we refer to them as meta-rules and meta-facts. As shown in Figure 2, to evaluate the practical inductive reasoning capability of models, we apply these metadata to various scenarios to generate concrete problems. Specifically, we have: **(1) List Transformation (LT):** List transformation is the primary format used in previous inductive reasoning tasks (Rule, 2020; Xu et al., 2024a; Chollet, 2019), and here we adopt this approach as well. We transform all fact vectors into one-dimensional lists and require the model to inductively infer the transformation rules applied to these lists. **(2) Real-world Problem (RP):** Previous datasets lack tests for inductive reasoning capabilities in real-world scenarios (Rule, 2020; Xu et al., 2024a; Qiu et al., 2024).[2] To mitigate this gap, we populate the metadata into different natural language templates across five real-life scenarios. The example in Figure 2 describes a trading scenario, where we use different items to represent different dimensions of the vector. All item transactions follow the same rule. **(3) Code Generation (CG):** For each fact, we use $x$ as the input and $y$ as the output of a function. The model is then tasked with predicting the corresponding Python function. **(4) String Transformation (ST):** The former three scenarios are related to numbers. Here, we replace the basic elements in the fact vectors with characters to conduct a new test. Notably, we modify the operations as follows: addition in the Add and Map operations is replaced with string concatenation, multiplication in Map is replaced with character replication, zero-padding in Pad becomes character deletion, and the numbers 0-9 are replaced with the characters $a$-$j$.

For humans, although the process of reasoning tends to be implicit, according to the traditional definition in logic, it can be divided into two stages: induction and deduction. Here, we focus on the objectives of these two stages: deriving the correct rules through induction and making inferences on new instances using deduction. We aim to explore the correlation between the two during the reasoning process of LLMs. Therefore, for each scenario, we design two tasks: rule induction (RI) and example inference (EI), defined as follows:

- **Rule Induction Task:** Given an observed fact set $\mathbb{X}$, this task evaluates the model's **accuracy** in inducing transformation rules $f$. It aims to evaluate the model's proficiency in mastering intermediate rules during inductive reasoning (Yang et al., 2024b; Shao et al., 2024).

- **Example Inference Task:** Given an observed fact set $\mathbb{X}$, the task provides an unseen example input $x_t$ as input and measures the **accuracy** of the predicted $y_t$. It aims to evaluate the final performance of the model's inductive reasoning (Chollet, 2019; Xu et al., 2024a; Qiu et al., 2024).

We provide all the prompts used for these tasks in Appendix A.3.

## 3 LANGUAGE MODELS ARE POOR RULE-BASED REASONERS

### 3.1 OVERALL PERFORMANCES ON MIRAGE

**Setup** We first evaluate the overall performance of various LLMs on MIRAGE. Here, we select GPT-4 (OpenAI, 2023), GPT-4o, Claude-3.5, Llama3-8B (Dubey et al., 2024), and Llama2-13B (Touvron et al., 2023) as representative models.[3] For the first three models, given their strong instruction-following capabilities, we provide only the instruction and allow them to answer the questions in a zero-shot setting. For the latter two models, to improve the format accuracy of the response, we additionally provide five examples before they answer the questions. Unless otherwise specified, we continue to use this setup to prompt the model in the subsequent experiments. For the dataset setting, we fix the size $N$ at 5 and measure performance across four

---

[2]Here, real-world scenarios refer to mathematical inductive reasoning within natural language contexts.

[3]Due to the frequency limitations of API calls, we can not conduct our evaluation on the latest o1 model.

scenarios when the dimension $D = 3, 5, 8$. We sample 500 questions for each test. More implementation details can be found in Appendix B.1.

| Model | Task | D=3 | | | | D=5 | | | | D=8 | | | |
|-------|------|-----|-----|-----|-----|-----|-----|-----|-----|-----|-----|-----|-----|
| | | LT | RP | CG | ST | LT | RP | CG | ST | LT | RP | CG | ST |
| Llama2-13B | RI | 0.01 | 0.00 | 0.00 | 0.03 | 0.01 | 0.01 | 0.00 | 0.21 | 0.00 | 0.01 | 0.00 | 0.10 |
| | EI | 0.26 | 0.11 | 0.25 | 0.22 | 0.13 | 0.03 | 0.14 | 0.25 | 0.06 | 0.01 | 0.06 | 0.19 |
| Llama3-8B | RI | 0.15 | 0.11 | 0.19 | 0.19 | 0.23 | 0.04 | 0.14 | 0.22 | 0.16 | 0.02 | 0.08 | 0.21 |
| | EI | 0.30 | 0.15 | 0.25 | 0.25 | 0.20 | 0.12 | 0.25 | 0.29 | 0.09 | 0.11 | 0.16 | 0.24 |
| GPT-4o | RI | 0.41 | 0.32 | 0.38 | 0.32 | 0.35 | 0.21 | 0.44 | 0.30 | 0.33 | **0.16** | 0.41 | 0.24 |
| | EI | 0.68 | 0.37 | 0.61 | 0.56 | 0.58 | 0.25 | 0.64 | 0.39 | 0.42 | 0.17 | 0.49 | 0.29 |
| GPT-4 | RI | **0.47** | 0.29 | **0.41** | 0.28 | **0.58** | **0.22** | **0.56** | 0.27 | **0.46** | 0.15 | **0.45** | 0.23 |
| | EI | 0.68 | 0.37 | 0.61 | 0.57 | 0.63 | 0.29 | 0.71 | 0.44 | 0.42 | 0.21 | 0.64 | 0.30 |
| Claude-3.5 | RI | 0.44 | **0.35** | 0.34 | **0.46** | 0.22 | 0.20 | 0.38 | **0.33** | 0.24 | 0.13 | 0.38 | **0.26** |
| | EI | 0.79 | 0.45 | 0.62 | 0.58 | 0.65 | 0.33 | 0.76 | 0.45 | 0.46 | 0.24 | 0.59 | 0.30 |

Table 1: Overall performance (accuracy) of different models on MIRAGE. The best results in rule induction (RI) are in **bold**, while the best results in example inference (EI) are underlined.

**Results** The results are shown in Table 1, from which we can draw the following conclusions: **(1) LLMs' inductive reasoning does not rely on rule induction.** Given the same set of observed facts, the model's performance on rule induction is noticeably worse than on example inference in almost all cases. This suggests that most of the model's correct deductions do not depend on inducing a correct rule. **(2) LLMs face difficulties in handling inductive reasoning in real-world problems.** When comparing different scenarios, all models perform the worst on the RP tasks. For example, GPT-4o only achieves **0.16** and **0.17** accuracy when the dimension is 8. This indicates that, compared to purely symbolic forms (LT, CG, ST), natural language forms pose a greater challenge for the models' inductive reasoning abilities.

**Supplementary Experiments** In the main experiments, we find that there is a significant performance gap between rule induction and example inference for LLMs. However, this gap may be caused by the difference in difficulty between the two tasks. When the model is unable to perform correct inductive reasoning, it is likely to guess the correct answers for the EI task more easily compared to the RI task, resulting in a higher accuracy. We conduct this additional experiment to eliminate the interference of this factor. Specifically, we randomly perturb one fact in $\mathbb{X}$ to violate rule $f$. Then, we observe the performance of both tasks and calculate the change rate (CR) of accuracy before and after the perturbation. CR represents the sensitivity of the model's performance to the input. If CR

| Model | Rule Induction | | | Example Inference | | |
|-------|------|------|------|------|------|------|
| | BF | AF | CR | BF | AF | CR |
| GPT-4o | 0.50 | 0.13 | 0.74 | 0.66 | 0.15 | 0.77 |
| Claude-3.5 | 0.37 | 0.07 | 0.81 | 0.65 | 0.22 | 0.66 |

Table 2: Comparison of CR on two tasks ($D = 3$, $N = 3$). BF and AF indicate the accuracy before and after perturbation.

is high, it indicates a strong correlation between task performance and input, making it difficult to answer the question correctly through random guessing, Therefore, CR can serve as an indicator of the reasoning difficulty for the task. We randomly choose 100 pieces of test data from the dataset and generate questions under the LF scenario.[4] The experimental results on different models are demonstrated in Table 2. We can observe that the two tasks have comparable CR for both models, indicating that the reasoning difficulty of the EI task is not lower than the RI task. The tasks themselves do not cause such a large performance gap.

## 3.2 PERFORMANCES OF ADVANCED METHODS

In §3.1, we observe that LLMs perform poorly on our dataset, especially in rule induction tasks. Considering previous work has proposed numerous methods to elicit the model's reasoning abilities (Wei et al., 2022; Wang et al., 2023b; Madaan et al., 2023), we wonder whether they can boost models' performance on MIRAGE.

**Setup** Since we focus on exploring the model's intrinsic capabilities, we only consider methods that do not introduce any external tools or knowledge. Specifically, the methods are as follows: **Input-Output (IO):** We prompt models to generate answers directly under different shots. **Inductive-Deductive (ID):** We prompt

---

[4]Unless otherwise specified, this configuration will be maintained for all subsequent experiments.

| Method | LT | | | RP | | | CG | | | ST | | |
|---|---|---|---|---|---|---|---|---|---|---|---|---|
| | RI | EI | (Δ) | RI | EI | (Δ) | RI | EI | (Δ) | RI | EI | (Δ) |
| IO (0-shot) | 0.46 | 0.76 | **0.30** | 0.43 | 0.72 | **0.28** | 0.39 | 0.46 | 0.08 | 0.47 | 0.70 | 0.23 |
| IO (5-shot) | 0.63 | 0.76 | 0.13 | **0.59** | 0.77 | 0.17 | **0.55** | 0.54 | -0.02 | 0.52 | **0.78** | 0.26 |
| ID (0-shot) | 0.46 | 0.56 | 0.11 | 0.46 | 0.57 | 0.11 | 0.33 | 0.42 | 0.09 | 0.22 | 0.65 | **0.43** |
| ID (5-shot) | 0.59 | 0.68 | 0.08 | 0.57 | 0.66 | 0.08 | 0.47 | 0.54 | 0.08 | 0.48 | 0.69 | 0.21 |
| CoT (0-shot) | 0.50 | 0.57 | 0.07 | 0.47 | 0.55 | 0.08 | 0.34 | 0.39 | 0.05 | 0.52 | 0.62 | 0.10 |
| CoT (5-shot) | 0.56 | 0.59 | 0.04 | 0.41 | 0.55 | 0.13 | 0.37 | 0.40 | 0.03 | 0.45 | 0.64 | 0.20 |
| SC (n=5) | 0.59 | 0.74 | 0.15 | 0.49 | 0.62 | 0.14 | 0.38 | 0.45 | 0.07 | **0.57** | 0.68 | 0.10 |
| SR (t=3) | 0.48 | 0.64 | 0.16 | 0.42 | 0.67 | 0.25 | 0.36 | 0.49 | **0.13** | 0.53 | 0.67 | 0.14 |
| HR (t=3, n=1) | 0.56 | 0.68 | 0.12 | 0.45 | 0.71 | 0.26 | 0.41 | 0.53 | 0.11 | 0.43 | 0.71 | 0.27 |
| HR (t=3, n=5) | **0.66** | **0.79** | 0.13 | **0.59** | **0.80** | 0.21 | **0.55** | **0.60** | 0.05 | 0.49 | 0.67 | 0.18 |

Table 3: Performance of different methods on MIRAGE using GPT-4o. The best results in each column are highlighted in **bold**, while the second best results are underlined.

models to generate rules for RI and apply them to answer questions in EI. **Chain-of-Thought (CoT)** (Wei et al., 2022): We prompt models to generate rationales and answers for the two tasks. **Self-Consistency (SC)** (Wang et al., 2023b): Based on CoT, we sample $n$ rationales and use the major voting strategy to predict the final answer. **Self-Refine (SR)** (Madaan et al., 2023): We prompt the model to provide feedback on the generated rules, and then refine the rules based on that feedback (with a maximum of $t$ iterations). After the iteration stops, we use the latest rule to answer the RI and apply it to answer the EI. **Hypothesis Refinement (HR)** (Qiu et al., 2024): HR is an optimized version of SR, which first generates $n$ rules. In each iteration, we apply the current rules to all observed examples, compare the actual output with the expected output, and get the number of correct predictions along with the information about incorrect examples. If a candidate rule is correct for all observed facts, it is immediately returned. Otherwise, the rule with the highest number of correct predictions and the associated error information is used as input for the model to refine, generating $n$ rules for the next round, until the maximum number of iterations $t$ is reached. We sample 200 questions for each test.

**Results** We illustrate the main results on GPT-4o in Table 3 (more results in Appendix B.2), from which we can conclude that: **(1) Advanced methods provide limited improvement to the model's inductive reasoning ability or may even have negative effects.** For both tasks, directly answering with few-shot settings can consistently achieve the highest or second-highest accuracy in most cases. After applying methods like CoT, the model's accuracy decreases by up to **18%** and **22%** on two tasks, respectively. It indicates that the key to optimizing inductive reasoning does not lie in refining the intermediate inductive process (as CoT-like methods do). **(2) The model's disregard for abstract rules during inductive reasoning is method-agnostic.** Although some methods use instructions to guide the model to focus more on the induced rules during reasoning (e.g. ID, SR, HR), there remains a significant gap in the model's RI and EI performance. For example, in the case of SR, the model's example inference accuracy outperforms its rule induction accuracy by an average of **16%**.

## 3.3 IMPACT OF INCREASING FACT SIZE

In the previous experiments, we consistently fix the observed fact numbers $N$. Therefore, as a supplement, we explore the impact of $N$ on the model's inductive reasoning process in this section. Theoretically, as the number of observed facts increases, the scope of the candidate rules narrows, which can lead to the incorrect inductive process becoming correct. If the reasoning process is rule-based, the model is likely to generate the correct rule (inductive) before applying it correctly (deductive). In other words, the time when the LLM induces the correct rule is no later than the time it performs the correct deduction. Thus, the cumulative number of observations required for the inductive rule to change from incorrect to correct should not exceed the number required for the test case to become correct. We design this experiment to validate whether it holds on the LLM.

**Setup** Given a fact set $\mathbb{X}^k$ of size $k$, and a fixed test input $\boldsymbol{x}_t$, we define the inductive correction threshold (ICT) and deductive correction threshold (DCT) as follows:

$$\text{ICT} = k \iff \forall i < k, \mathcal{M}(\mathbb{X}^i|I) \neq f \wedge \mathcal{M}(\mathbb{X}^k|I) = f \tag{3}$$

$$\text{DCT} = k \iff \forall i < k, \mathcal{M}(\mathbb{X}^i, \boldsymbol{x}_t|D) \neq f(\boldsymbol{x}_t) \wedge \mathcal{M}(\mathbb{X}^k, \boldsymbol{x}_t|D) = f(\boldsymbol{x}_t) \tag{4}$$

Here, $\mathcal{M}(\cdot|I), \mathcal{M}(\cdot|D)$ are the model's outputs in RI and EI tasks. We set $D = 5$ and vary $N$, then analyze the distribution of these two thresholds across 100 samples, reporting the results in Figure 3.

**Results** Based on the result, we further demonstrate that **LLM's deduction does not rely on an inductive rule**. From both of the two figures, we can observe that most points are distributed in the upper left region

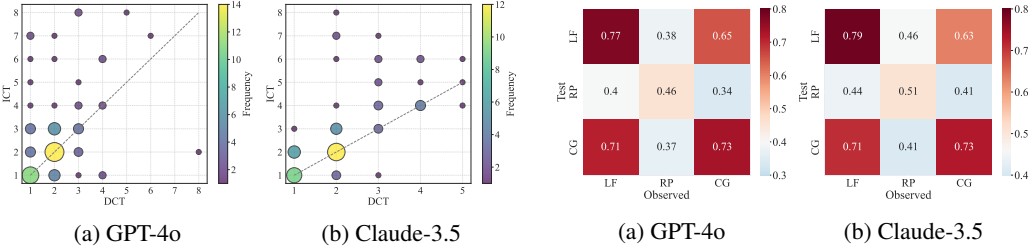

Figure 3: The distribution of ICT and DCT for the examples across different models.

Figure 4: Performance on EI tasks under different scenarios of observed and test facts.

of the line $x = y$, indicating that for the vast majority of cases, DCT is smaller than ICT. Therefore, the fact numbers $N$ does not affect the conclusion we stated earlier. LLM requires fewer facts to successfully perform an example inference task compared to correct induction.

## 3.4 TRANSFERABILITY TEST OF INDUCTIVE RULES

Finally, we investigate the impact of different scenarios on the inductive reasoning process. For rule-based reasoning, once a rule is formed through induction, it should be transferable. That is, a rule induced in one scenario should be applicable to another scenario with the same underlying transformation. We experiment to explore whether LLMs possess this ability when performing inductive reasoning.

**Setup** Specifically, we exclude ST in this experiment since its basic transformations differ from the other three scenarios (see §2.1). For the remaining three scenarios, we generate the observed facts in one scenario, and then transform the test case into another scenario. Since our dataset can generate questions in different scenarios based on the same meta-rule, we can easily ensure that they share the same underlying transformation.

**Results** From the results shown in Figure 4, we can get that: **(1) LLMs lack transferability in inductive reasoning.** Across different cases, the highest performance occurs when the scenarios of the observed and test facts are consistent (i.e., the diagonal from the top left to the bottom right in the figure).**(2) The inductive reasoning process of the LLM is form-related.** Compared to the transfer between LT and RP (or CG and RP), the transfer between LT and CG demonstrates better performance. We infer that this is because the forms of LT and CG are more similar (see Figure 2). In addition, we also design experiments under the fine-tuning paradigm to compare the model's transferability (presented in Appendix B.3), and the results remain consistent. Based on the above two observations, we further confirm that LLMs do not rely on abstract rules when performing inductive reasoning. So, what is the underlying mechanism behind it? In the following section, we focus on addressing this question.

## 4 LANGUAGE MODELS ARE GOOD NEIGHBOR-BASED REASONERS

### 4.1 MOTIVATION

From § 3.4, we know that closer forms between the observed facts and the test case can enable the model to perform inductive reasoning more effectively. However, is the positive impact brought by the similarity limited only to the form? The answer to this question is likely "No". Upon reviewing related works, we find that models tend to match various similar patterns in the context and use them to predict the next token (Olsson et al., 2022; Wang et al., 2023a; Hu et al., 2024b). Therefore, we aim to identify a metric to measure some other similarities between the observed facts and the test input. Since all of our facts are transformed from vectors, we associate this similarity with the distance between these facts in feature space.

In topology, if $f : X \rightarrow Y$ is a continuous function between two Euclidean spaces and $\mathcal{N}(\boldsymbol{x}_0, \epsilon)$ is a $\epsilon$-neighborhood of the point $\boldsymbol{x}_0$ in $X$, then we have:

$$\exists \eta > 0, \text{ s.t. } \forall \boldsymbol{x} \in \mathcal{N}(\boldsymbol{x}_0, \epsilon), \ f(\boldsymbol{x}) \in \mathcal{N}(f(\boldsymbol{x}_0), \eta) \tag{5}$$

In other words, continuous functions preserve the neighborhood property. If a fact input vector $\boldsymbol{x}$ closes to the test input $\boldsymbol{x}_t$, then their output vectors $\boldsymbol{y}$ and $\boldsymbol{y}_t$ will remain close.[5] Therefore, **the close distance between $\boldsymbol{y}_t$ and $\boldsymbol{y}$ may allow LLM to predict $\boldsymbol{y}_t$ based on $\boldsymbol{y}$ in observed facts without the need for correct rule generation**. In the following sections, we demonstrate through experiments that the model's inductive reasoning relies on this paradigm, which we refer to as **neighbor-based reasoning**.

---

[5]We rigorously prove in Appendix C.1 that the rules $f$ in our dataset are all continuous functions.

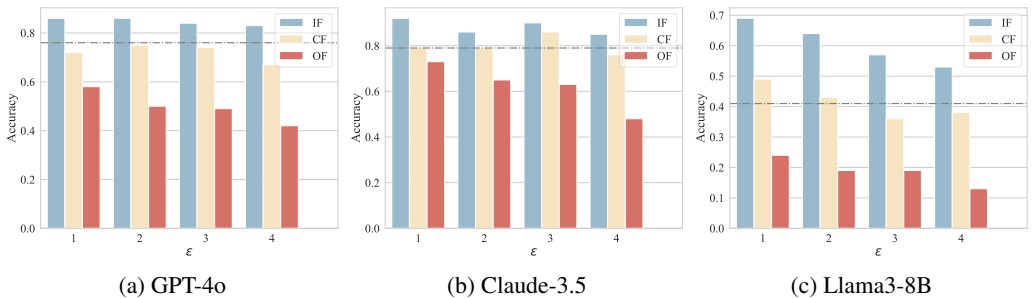

Figure 5: Performance of different fact types on our dataset ($D = 3$, $N = 5$). The dashed line represents the baseline accuracy.

| Type | N=3 | | | | N=5 | | | | N=8 | | | |
|---|---|---|---|---|---|---|---|---|---|---|---|---|
| | LT | RP | CG | ST | LT | RP | CG | ST | LT | RP | CG | ST |
| Baseline | 0.52 | 0.19 | 0.78 | 0.42 | 0.66 | 0.36 | 0.71 | 0.46 | 0.76 | 0.34 | 0.80 | 0.54 |
| IF Only | **0.78** | **0.46** | **0.82** | **0.59** | **0.84** | **0.52** | **0.84** | **0.63** | **0.86** | **0.54** | **0.91** | **0.70** |
| CF Only | 0.48 | 0.25 | 0.59 | 0.43 | 0.69 | 0.35 | 0.72 | 0.50 | 0.75 | 0.34 | 0.82 | 0.53 |
| OF Only | 0.46 | 0.18 | 0.50 | 0.38 | 0.49 | 0.23 | 0.57 | 0.36 | 0.61 | 0.23 | 0.67 | 0.38 |

Table 4: Performance of different fact types under various settings ($D = 5$). The best results in each column are highlighted in **bold**, while the worst results are underlined.

## 4.2 NEIGHBOR FACTS IN INDUCTIVE REASONING

Before conducting the experiments, we first define some key concepts in our work: the distance $d$ and neighborhood $\mathcal{N}$. In our setup, the components at corresponding positions in the vectors follow the same transformation rules, while non-corresponding components may undergo different transformations (see Equation 2). Hence, we consider using the distance based on the corresponding components: Chebyshev distance (further discussion in Appendix C.2). Given observed fact $\mathbb{X}_i = (\boldsymbol{x}_i, \boldsymbol{y}_i)$ and test input $\boldsymbol{x}_t$, we have:

$$d(\mathbb{X}_i, \boldsymbol{x}_t) = \max_k \left( |x_{ik} - x_{tk}| \right) \tag{6}$$

where $x_{ik}$ and $x_{tk}$ are the $k$-th component of two input vectors. Then we can define the $\epsilon$-neighborhood of $\boldsymbol{x}_t$ based on the distance:

$$\mathcal{N}(\boldsymbol{x}_t, \epsilon) = \{\mathbb{X}_i \mid d(\boldsymbol{x}_i - \boldsymbol{x}_t) \le \epsilon\} \tag{7}$$

**Setup** According above definitions, we can divide an observed fact $\mathbb{X}_k$ into three categories based on the distance between $\boldsymbol{x}$ and the test input $\boldsymbol{x}_t$: (1) **In-neighborhood Fact (IF):** If $\mathbb{X}_k \in \mathcal{N}(\boldsymbol{x}_t, \epsilon)$, we call $\mathbb{X}_k$ is a in-neighborhood fact. (2) **Cross-neighborhood Fact (CF):** If $\mathbb{X}_k \notin \mathcal{N}(\boldsymbol{x}_t, \epsilon)$, but $\exists i \in [1, D]$, s.t. $|x_i - x_{ti}| \le \epsilon$, we consider it a suboptimal neighbor fact because some of its components can still contribute to the model's inductive reasoning process. In this case, we call $\mathbb{X}_k$ is a cross-neighborhood fact. (3) **Out-neighborhood Fact (OF):** If $\forall i \in [1, D]$, $|x_i - x_{ti}| > \epsilon$, we call $\mathbb{X}_k$ is an out-neighborhood fact. By generating examples based on these definitions, we can make the fact set $\mathbb{X}$ contain only one type of fact. After constructing different fact sets, we compare the model's performance on EI tasks under these settings. Besides, we use the performance under the default fact set as the baseline, where all facts are randomly sampled without any constraints.

**Results** We report the results in Figure 5. It demonstrates that: **(1) LLM's inductive reasoning is neighbor-based.** By comparing three settings, we find that observed facts closer to the test case result in better performance (IF > CF > OF) across all models. Besides, compared to the baseline (the dashed line in figures), the accuracy significantly drops after removing the neighbor cases in $\mathbb{X}$ (i.e. OF). These phenomena indicate that the model heavily relies on neighbor facts during reasoning. **(2) LLMs have a strong ability to capture neighboring patterns.** When we set the neighborhood radius $\epsilon$ to 4, both IF and CF still contribute to high accuracy for the model. Besides, OF continues to show a significant decline (compared to $\epsilon = 3$). These observations indicate that LLMs can still learn similar patterns even when the observed facts are relatively distant.

## 4.3 UNIVERSALITY OF NEIGHBOR-BASED REASONING

We consider whether LLM's inductive reasoning universally relies on neighbor cases, hence, we set $\epsilon$ to 1 and repeat the experiment under different settings, where the baseline is the same as § 4.2. The results on GPT-

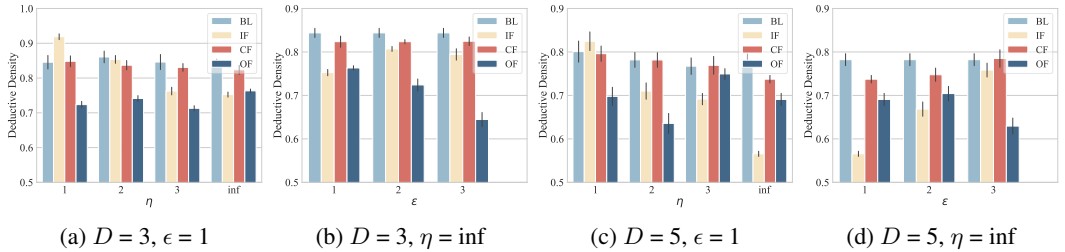

| (a) $D = 3, \epsilon = 1$ | (b) $D = 3, \eta = \inf$ | (c) $D = 5, \epsilon = 1$ | (d) $D = 5, \eta = \inf$ |

Figure 7: Deductive Density ($I_d$) of various fact types on GPT-4o under different test radius $\eta$ and neighborhood radius $\epsilon$ ($N = 5$). BL represents the performance of the baseline, where we use default fact sets with no substitution.

4o are reported in Table 4, which demonstrates that the **neighbor-based paradigm is universal in LLMs' inductive reasoning process**. Across different scenarios and fact numbers, IF consistently gets the highest accuracy, while OF gets the lowest accuracy. The reliance of LLMs' inductive reasoning on neighbor facts is independent of the specific task scenarios, models, or fact numbers. We present more results in Appendix C.3.

## 4.4 EFFECTIVE SCOPE OF NEIGHBOR-BASED REASONING

We have demonstrated that neighbor examples in observed facts significantly affect the model's performance on the test case. However, what is the effective scope of it? Is it only pattern matching on a single test example or reasoning an implicit rule that affects more examples? To answer the question, we first make three assumptions about its possible scope and show them in Figure 6. For individual scope, the model can only answer the test case $\boldsymbol{x}_t$ (e.g. [3,4,7] in the figure), for all other cases, the accuracy of the prediction is very low. For localized scope, the model can also answer cases close to $\boldsymbol{x}_t$ (i.e. the neighbor facts of $\boldsymbol{x}_t$). For global scope, the model can answer all cases with high accuracy.

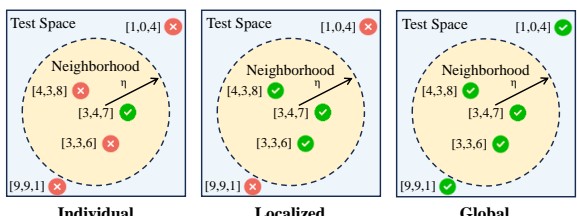

Figure 6: Examples for three different effective scopes.

**Setup** In our experiment, we sample $n$ test cases $\mathbb{X}_t$ (here $n = 5$) in each EI task $\tau$ and define the accuracy $a_\tau$ for this particular task as:

$$a_\tau = \frac{1}{n} \sum_{\boldsymbol{x} \in \mathbb{X}_t} \mathbb{I}[\mathcal{M}(\mathbb{X}_\tau, \boldsymbol{x}|D) = f(\boldsymbol{x})] \tag{8}$$

Here $\mathbb{X}_\tau$ is the observed fact set of the task. Let $T$ denote the set of all EI tasks (we set $|T| = 100$), we define **deductive density** $I_d$ as:

$$I_d = \frac{1}{|T_c|} \sum_{\tau \in T} a_\tau \mathbb{I}[\mathcal{M}(\mathbb{X}_\tau, \boldsymbol{x}_t|D) = f(\boldsymbol{x}_t)] \tag{9}$$

$$|T_c| = \sum_{\tau \in T} \mathbb{I}[\mathcal{M}(\mathbb{X}_\tau, \boldsymbol{x}_t|D) = f(\boldsymbol{x}_t)] \tag{10}$$

where $\boldsymbol{x}_t$ is the origin test input in task $\tau$. We use this metric to indicate the impact of a successful deduction (i.e. [3, 4, 7] in Figure 6) on reasoning over other examples in the test region $\mathcal{N}(\boldsymbol{x}_t, \eta)$. A high $I_d$ indicates that the model performs well in most cases within this region, while a low $I_d$ suggests that the model's reasoning is more localized or even individual. For comparison, we set the test radius $\eta$ to 1, 2, 3, and infinity (i.e. the full test space), and calculate the corresponding $I_d$ for the model. Besides, we also vary the neighborhood radius $\epsilon$ to examine the impact of different distributions of neighbor facts on their effective scope (here we set the test region to the full space). We repeat the experiment five times to eliminate the interference of random errors, and the results are illustrated in Figure 7.

**Results** We can draw conclusions as follows: **(1) LLM conducts localized reasoning through the neighbor-based paradigm.** From Figure 7a, 7c, we observe that the $I_d$ of IF and CF decreases continuously as the radius of the test domain expands. These neighbor cases are highly effective within the neighborhood of

$\boldsymbol{x}_t$. For example, in Figure 7a, when $\eta = 1$, the model can achieve over 0.9 $I_d$. However, this impact diminishes for test cases that are farther from $\boldsymbol{x}_t$. As an example, in Figure 7c, the model only gets around 0.5 $I_d$ in full test space. **(2) The effective scope of neighbor facts is proportional to their distance from the test case.** According to Figure 7b,7d, the $I_d$ of IF and CF (particularly IF) increases as $\epsilon$ becomes large. When the neighborhood radius increases, the distribution of these facts becomes more dispersed. We can infer that a more dispersed distribution of neighbor facts tends to make the effective scope more global.

## 5 LIMITATIONS AND DISCUSSIONS

**Interpretation Methods**   Most model interpretation studies delve into the internal of models (e.g. neurons, attention layers), providing a comprehensive explanation of the working mechanisms (Romera-Paredes et al., 2024; Li et al., 2024). However, our work does not conduct internal analysis but instead relies on performance comparisons under different settings. There are two main considerations for this: On one hand, this work aims to identify mechanisms that are applicable to black-box models. Since we do not have access to the internal parameters of these models, we are unable to use previous methods. On the other hand, white-box models exhibit poor inductive reasoning capabilities according to Table 1. Therefore, conducting in-depth interpretations based on white-box models may introduce noise to the conclusions.

**Experimental Settings**   The goal of this paper is to evaluate and explain the inductive reasoning process in LLMs, rather than to improve the task performance. Therefore, we do not meticulously design the prompts used in the experiments, nor do we use the best-performing inductive reasoning methods throughout the analysis. We believe that the experimental setup of 0-shot IO with simple instructions is more aligned with real-world application scenarios, making our evaluation and explanation results more meaningful.

**Future Directions**   Our study demonstrates that LLMs perform poorly in rule-based reasoning but excel at using neighbor facts for reasoning. Future work could explore methods to encourage the model to follow rules more closely during reasoning or to further optimize the model's inductive reasoning abilities based on this neighbor-matching finding.

## 6 RELATED WORK

**Evaluating Inductive Reasoning Abilities of LLMs.**   Existing studies on evaluating LLM's inductive reasoning capabilities mainly use only a single task. On one hand, some works assess the model's rule induction ability by evaluating the inference accuracy on unseen examples (Moskvichev et al., 2023; Tang et al., 2023; Gendron et al., 2023; Xu et al., 2024b; Qiu et al., 2024). However, since the model's deduction does not always rely on inducing the correct rule, this indirect evaluation method can introduce some inaccuracies. On the other hand, some studies directly evaluate the correctness of the generated rules to assess inductive reasoning ability (Shao et al., 2024; Cheng et al., 2024; Yang et al., 2024b; Wang et al., 2025). These studies lack evaluation on test examples, making it difficult to confirm the model's mastery of the inductive rules. Our work evaluates both aspects, providing a comprehensive analysis of the model's inductive reasoning process.

**Mechanism Analysis on LLM's Reasoning.**   A growing body of interpretability research has begun analyzing the reasoning mechanisms of LLMs, aiming to deepen our understanding of how these models function. Some studies explore the mechanisms behind mathematical reasoning (Zhang et al., 2024; Hu et al., 2024b; Romera-Paredes et al., 2024; Stolfo et al., 2023), some works investigate multi-hop reasoning (Wang et al., 2024a; Hou et al., 2023; Yang et al., 2024a; Biran et al., 2024), and some focus on other types of reasoning (Li et al., 2024; Hu et al., 2024a). However, there is currently a lack of analysis on the mechanisms of inductive reasoning. Our work mitigates this gap and uncovers the neighbor-based paradigms LLMs follow when performing inductive reasoning.

## 7 CONCLUSION

In this paper, we focus on evaluating and explaining the inductive reasoning process of LLMs. First, we construct a dataset MIRAGE, which provides both inductive and deductive evaluation tasks, with the flexibility to generate test examples in any distribution, different difficulties, and various forms. Based on it, we demonstrated that LLM is a poor rule-based reasoner, it does not need to rely on inductive rules when performing inductive reasoning. Compared to correct induction, the model can perform successful deduction with fewer observations, and this deduction is closely related to the form of the input. Furthermore, we identify a key paradigm of LLM inductive reasoning: neighbor-based reasoning. The model tends to leverage observed facts that are close to the test examples in feature space for inductive reasoning. Through it, the model can achieve strong inductive reasoning capabilities within a localized scope and apply this ability to make inferences on unseen examples.

ACKNOWLEDGMENTS

This work is supported by the National Natural Science Foundation of China (No.U24A20335, No. 62176257, No. 62406321). This work is also supported by Beijing Natural Science Foundation (L243006). This work is also supported by the Youth Innovation Promotion Association CAS and the China Postdoctoral Science Foundation under Grant Number 2024M753500.

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

## A    MORE DETAILS FOR DATASET CONSTRUCTION

### A.1    COMPARISON OF DATASETS BETWEEN RELATED WORK AND OUR STUDY

Since our work is conducted entirely on our MIRAGE dataset, we aim to provide a detailed comparison with representative datasets from related studies to demonstrate its effectiveness. Specifically, we report the comparison in Table 5. From the results, we can see that our dataset can cover most of the operations and forms in previous datasets. For example, the transformation examples in the 1D-ARC dataset shown in the table are equivalent to our PAD operation. Therefore, we demonstrate that our dataset is an effective dataset for inductive reasoning. Moreover, based on its coverage, conducting experiments solely on it is sufficient.

| Dataset | Fact Form | Main Operation | Example |
|---|---|---|---|
| ARC (Chollet, 2019) | List (2D) | Fill, Move, Pile | input: [[0,1,0],[1,1,0], [0,1,0],[0,1,1],[0,1,0], [1,1,0]]

output: [[0,2,0],[2,2,0], [0,2,0],[0,2,2],[0,2,0], [2,2,0],[0,2,0],[0,2,2], [0,2,0] |
| MiniSCAN (Lake et al., 2019) | String | String Translation | input: her sneury voirk

output: GREEN BLUE |
| ListFunctions (Rule, 2020) | List | List Operation | input: [4,7,6,9,0]

output:[4,8,6,9,0] |
| MiniARC (Kim et al., 2022) | List (2D) | Fill, Move, Pile | input: [[1,1,5,6,8], [0,1,5,6,6],[5,5,5,5,5], [7,7,5,4,4],[7,7,5,0,4]]

output: [[1,6,0,0,0], [7,4,0,0,0],[0,0,0,0,0], [0,0,0,0,0], [0,0,0,0,0]] |
| 1D-ARC (Xu et al., 2024a) | List | Fill, Move, Pile | input: [0,0,2,0,0,0,0,2, 0,0,0]

output: [0,0,2,2,2,2,2,2, 0,0,0] |
| Case2Code (Shao et al., 2024) | Code | Python Function | input: dict(no=2)

output: [2] |
| MIRAGE | All above + RP | All above | See Figure 2 |

Table 5: Comparison between some representative datasets and ours.

### A.2    EVALUATION OF MATHEMATICAL OPERATION DIFFICULTY IN MIRAGE

Our data introduces some mathematical operations in Add and Map, here we aim to demonstrate that these calculations are inherently simple for LLMs, ensuring that they do not interfere with our evaluation of the model's inductive reasoning performance. Specifically, we randomly construct linear operations in Add and Map with single-digit operands (cover all of the operations included in this paper) and observe the accuracy of each model on 100 questions. The results are reported in Table 6. We can observe that most of the modes can achieve very high accuracy on these math operations. Specifically, all closed-source models achieve 100% accuracy. This indicates that our dataset construction effectively eliminates noise introduced by mathematical calculations in most cases.

| Operation | Llama2-13B | Llama3-8B | GPT-4o | GPT-4 | Claude-3.5 |
|---|---|---|---|---|---|
| Map | 0.96 | 0.93 | 1.00 | 1.00 | 1.00 |
| Add | 0.51 | 0.99 | 1.00 | 1.00 | 1.00 |

Table 6: Accuracy of basic mathematical computations across different models in our dataset.

## A.3 EVALUATION TEMPLATES FOR DIFFERENT TASKS AND SCENARIOS

In Tables 14 and 15, we report the evaluation prompts used to evaluate the models in our work. In Table 16, we provide the templates used for constructing different scenarios in RP.

## B MORE DETAILS FOR RULE-BASED REASONING EVALUATION

### B.1 IMPLEMENT DETAILS FOR MAIN EXPERIMENTS

For model version, we select `Llama-2-13b-chat-hf`, `Meta-Llama-3-8B-Instruct`, `gpt-4-0613`, `gpt-4o-2024-05-13` and `claude-3-5-sonnet-20240620`. All experiments are conducted on 4 NVIDIA GeForce RTX 3090 GPUs. For the sake of simplicity, we include all the prompts used in this work in the supplementary materials.

### B.2 MORE EXPERIMENTS ON OTHER MODELS

In this section, we repeat the experiments in § 3.2 on the Llama3-8B model and Llama2-13B model, the results are shown in Table 7, 8. Here we set $D = 3$, $N = 5$. We can observe that the results in our main text still hold on these models. We do not apply HR on them, since these two models have difficulty in evaluating the rule based on given templates under 0-shot settings.

| Method | LT | | | RP | | | CG | | | ST | | |
|---|---|---|---|---|---|---|---|---|---|---|---|---|
| | RI | EI | ($\Delta$) | RI | EI | ($\Delta$) | RI | EI | ($\Delta$) | RI | EI | ($\Delta$) |
| IO (0-shot) | 0.21 | 0.54 | 0.33 | 0.03 | 0.48 | 0.45 | 0.17 | 0.22 | 0.05 | 0.07 | 0.45 | 0.38 |
| IO (5-shot) | 0.32 | 0.41 | 0.09 | 0.35 | 0.40 | 0.05 | 0.14 | 0.24 | 0.10 | 0.25 | 0.35 | 0.10 |
| ID (0-shot) | 0.10 | 0.24 | 0.14 | 0.12 | 0.30 | 0.18 | 0.10 | 0.17 | 0.07 | 0.01 | 0.35 | 0.34 |
| ID (5-shot) | 0.35 | 0.44 | 0.09 | 0.30 | 0.40 | 0.10 | 0.01 | 0.04 | 0.03 | 0.31 | 0.33 | 0.02 |
| CoT (0-shot) | 0.25 | 0.39 | 0.14 | 0.13 | 0.39 | 0.26 | 0.16 | 0.14 | -0.02 | 0.16 | 0.42 | 0.26 |
| CoT (5-shot) | 0.54 | 0.59 | 0.05 | 0.36 | 0.40 | 0.04 | 0.41 | 0.55 | 0.14 | 0.47 | 0.66 | 0.19 |
| SC (n=5) | 0.47 | 0.54 | 0.07 | 0.35 | 0.37 | 0.02 | 0.45 | 0.55 | 0.10 | 0.51 | 0.62 | 0.11 |
| SR (t=3) | 0.16 | 0.26 | 0.10 | 0.09 | 0.27 | 0.18 | 0.08 | 0.20 | 0.12 | 0.05 | 0.39 | 0.34 |

Table 7: Performance of different methods on MIRAGE using Llama3-8B (100 examples).

| Method | LT | | | RP | | | CG | | | ST | | |
|---|---|---|---|---|---|---|---|---|---|---|---|---|
| | RI | EI | ($\Delta$) | RI | EI | ($\Delta$) | RI | EI | ($\Delta$) | RI | EI | ($\Delta$) |
| IO (0-shot) | 0.01 | 0.29 | 0.28 | 0.01 | 0.41 | 0.40 | 0.01 | 0.15 | 0.15 | 0.18 | 0.40 | 0.22 |
| IO (5-shot) | 0.02 | 0.37 | 0.35 | 0.01 | 0.34 | 0.33 | 0.01 | 0.14 | 0.14 | 0.05 | 0.30 | 0.25 |
| ID (0-shot) | 0.00 | 0.02 | 0.02 | 0.00 | 0.03 | 0.03 | 0.00 | 0.06 | 0.06 | 0.17 | 0.20 | 0.02 |
| ID (5-shot) | 0.02 | 0.12 | 0.10 | 0.01 | 0.14 | 0.14 | 0.00 | 0.01 | 0.01 | 0.01 | 0.30 | 0.29 |
| CoT (0-shot) | 0.03 | 0.10 | 0.08 | 0.03 | 0.20 | 0.17 | 0.00 | 0.13 | 0.13 | 0.06 | 0.13 | 0.08 |
| CoT (5-shot) | 0.01 | 0.24 | 0.23 | 0.02 | 0.14 | 0.12 | 0.00 | 0.14 | 0.14 | 0.07 | 0.10 | 0.03 |
| SC (n=5) | 0.07 | 0.24 | 0.17 | 0.08 | 0.49 | 0.41 | 0.01 | 0.34 | 0.33 | 0.17 | 0.37 | 0.20 |
| SR (t=3) | 0.00 | 0.03 | 0.03 | 0.01 | 0.07 | 0.06 | 0.00 | 0.06 | 0.06 | 0.01 | 0.10 | 0.09 |

Table 8: Performance of different methods on MIRAGE using Llama2-13B (200 examples).

### B.3 EXPERIMENTS ON FINE-TUNING METHOD

In the main text, we primarily use in-context learning to evaluate the performance of various models. Here, we supplement the evaluation with the performance of fine-tuned models on MIRAGE. Specifically, we use `Meta-Llama-3-8B-Instruct` as the backbone, setting N = 5 and D = 5, and train the model on 8,000 samples. For the training parameters, we set the learning rate to 0.0001, the batch size to 1, and the number of epochs to 10. Additionally, LoRA is employed to train people on different types of tasks. The performances on 100 test examples are presented in Table 9. It demonstrates that fine-tuning can effectively enhance the model's inductive reasoning capabilities. Compared to the 5-shot ICL, the model performs better on both reasoning tasks, even when not trained on tasks of the same format. However, consistent with the conclusion in § 3.4, the results of the training do not exhibit good formal generalization. The model tends to perform relatively poorly on test sets with significantly different forms from the training set (e.g., LT and RP).

| Method | LT | | RP | | CG | | ST | |
|---|---|---|---|---|---|---|---|---|
| | RI | EI | RI | EI | RI | EI | RI | EI |
| 5-shot ICL | 0.31 | 0.27 | 0.04 | 0.17 | 0.16 | 0.28 | 0.21 | 0.32 |
| LT Training | **0.89** | **0.82** | 0.22 | 0.24 | 0.34 | 0.80 | 0.26 | **0.38** |
| RP Training | 0.51 | 0.44 | **0.78** | **0.74** | 0.42 | 0.69 | 0.24 | 0.35 |
| CG Training | 0.76 | 0.75 | 0.22 | 0.25 | **0.86** | **0.80** | 0.26 | 0.37 |
| ST Training | 0.51 | 0.52 | 0.19 | 0.21 | 0.33 | 0.61 | **0.50** | 0.35 |

Table 9: Performance of the fine-tuned model on MIRAGE. The best results in each column are highlighted in **bold**.

## C MORE DETAILS FOR NEIGHBOR-BASED REASONING EVALUATION

### C.1 PROOF OF CONTINUES FUNCTIONS

Here, we prove that the five basic vector operations in MIRAGE are all continuous functions:

**Theorem 1** (Add Operation Continuity). *Let $\mathbf{A} = (a_1, a_2, \ldots, a_n) \in \mathbb{R}^n$. Define a mapping $f : \mathbb{R}^n \to \mathbb{R}^n$ such that for a fixed index $k \in \{1, 2, \ldots, n\}$ and a fixed subset $I \subseteq \{1, 2, \ldots, n\}$, we have*

$$f(\mathbf{A}) = (a_1, \ldots, a_{k-1}, \sum_{i \in I} a_i, a_{k+1}, \ldots, a_n),$$

*where $k \notin I$. Then $f$ is a continuous function.*

*Proof.* Consider two vectors $\mathbf{A}, \mathbf{B} \in \mathbb{R}^n$:

$$\mathbf{A} = (a_1, a_2, \ldots, a_n), \quad \mathbf{B} = (b_1, b_2, \ldots, b_n).$$

The mapping $f$ replaces the $k$-th element of the vector with the sum of elements indexed by the subset $I$. Thus,

$$f(\mathbf{A}) = (a_1, \ldots, a_{k-1}, \sum_{i \in I} a_i, a_{k+1}, \ldots, a_n),$$

$$f(\mathbf{B}) = (b_1, \ldots, b_{k-1}, \sum_{i \in I} b_i, b_{k+1}, \ldots, b_n).$$

The distance between the images of $\mathbf{A}$ and $\mathbf{B}$ under $f$ is

$$\|f(\mathbf{A}) - f(\mathbf{B})\| = \sqrt{\sum_{j=1, j \neq k}^{n} (a_j - b_j)^2 + \left(\sum_{i \in I} a_i - \sum_{i \in I} b_i\right)^2}.$$

Let us focus on the term involving the sums:

$$\sum_{i \in I} a_i - \sum_{i \in I} b_i = \sum_{i \in I} (a_i - b_i).$$

By the triangle inequality, we have

$$\left| \sum_{i \in I} (a_i - b_i) \right| \le \sum_{i \in I} |a_i - b_i|.$$

Therefore,

$$\left( \sum_{i \in I} a_i - \sum_{i \in I} b_i \right)^2 \le \left( \sum_{i \in I} |a_i - b_i| \right)^2.$$

Using the Cauchy-Schwarz inequality, we get

$$\left( \sum_{i \in I} |a_i - b_i| \right)^2 \le |I| \sum_{i \in I} (a_i - b_i)^2,$$

where $|I|$ is the cardinality of the set $I$.

Therefore,

$$\|f(\mathbf{A}) - f(\mathbf{B})\| \le \sqrt{\sum_{j=1, j \neq k}^{n} (a_j - b_j)^2 + |I| \sum_{i \in I} (a_i - b_i)^2}.$$

This can be bounded as

$$\|f(\mathbf{A}) - f(\mathbf{B})\| \le C \|\mathbf{A} - \mathbf{B}\|,$$

where $C$ is a constant depending on $n$ and $|I|$.

Therefore, for any $\epsilon > 0$, choose $\delta = \frac{\epsilon}{C}$. If $\|\mathbf{A} - \mathbf{B}\| < \delta$, then

$$\|f(\mathbf{A}) - f(\mathbf{B})\| < C\delta = \epsilon.$$

Hence, $f$ is continuous. $\qquad\square$

**Theorem 2** (Copy Operation Continuity). *Let* $\mathbf{A} = (a_1, a_2, \ldots, a_n) \in \mathbb{R}^n$. *Define a mapping* $f : \mathbb{R}^n \to \mathbb{R}^n$ *such that for fixed indices* $J \subseteq \{1, 2, \ldots, n\}$ *and a fixed index* $k \in \{1, 2, \ldots, n\}$, *we have*

$$f(\mathbf{A}) = (b_1, b_2, \ldots, b_n),$$

*where*

$$b_i = \begin{cases} a_k & \text{if } i \in J, \\ a_i & \text{otherwise.} \end{cases}$$

*Then* $f$ *is a continuous function.*

*Proof.* Consider two vectors $\mathbf{A}, \mathbf{B} \in \mathbb{R}^n$:

$$\mathbf{A} = (a_1, a_2, \ldots, a_n), \quad \mathbf{B} = (b_1, b_2, \ldots, b_n).$$

The mapping $f$ replaces each element of $\mathbf{A}$ at the positions indexed by $J$ with the value of the element at index $k$. Specifically,

$$f(\mathbf{A}) = (c_1, c_2, \ldots, c_n),$$

where

$$c_i = \begin{cases} a_k & \text{if } i \in J, \\ a_i & \text{otherwise.} \end{cases}$$

Similarly,

$$f(\mathbf{B}) = (d_1, d_2, \ldots, d_n),$$

where

$$d_i = \begin{cases} b_k & \text{if } i \in J, \\ b_i & \text{otherwise.} \end{cases}$$

The distance between the images of $\mathbf{A}$ and $\mathbf{B}$ under $f$ is given by

$$\|f(\mathbf{A}) - f(\mathbf{B})\| = \sqrt{\sum_{i=1}^{n} (c_i - d_i)^2}.$$

By the definition of $f$, we have

$$c_i - d_i = \begin{cases} a_k - b_k & \text{if } i \in J, \\ a_i - b_i & \text{otherwise.} \end{cases}$$

Therefore, the distance can be rewritten as

$$\|f(\mathbf{A}) - f(\mathbf{B})\| = \sqrt{\sum_{i \in J}(a_k - b_k)^2 + \sum_{i \notin J}(a_i - b_i)^2}.$$

Since the sum over $J$ has $|J|$ terms that are all equal to $(a_k - b_k)^2$, this simplifies to

$$\|f(\mathbf{A}) - f(\mathbf{B})\| = \sqrt{|J|(a_k - b_k)^2 + \sum_{i \notin J}(a_i - b_i)^2}.$$

This can be bounded as

$$\|f(\mathbf{A}) - f(\mathbf{B})\| \leq C\|\mathbf{A} - \mathbf{B}\|,$$

where $C$ is a constant depending on $n$ and $|J|$.

Therefore, for any $\epsilon > 0$, choose $\delta = \frac{\epsilon}{C}$. If $\|\mathbf{A} - \mathbf{B}\| < \delta$, then

$$\|f(\mathbf{A}) - f(\mathbf{B})\| < C\delta = \epsilon.$$

Hence, $f$ is continuous. $\qquad\square$

**Theorem 3** (Map Operation Continuity). *Let $\mathbf{A} = (a_1, a_2, \ldots, a_n) \in \mathbb{R}^n$. Define a mapping $f : \mathbb{R}^n \to \mathbb{R}^n$ such that for fixed indices $J \subseteq \{1, 2, \ldots, n\}$ and fixed scalars $k, b \in \mathbb{R}$, we have*

$$f(\mathbf{A}) = (b_1, b_2, \ldots, b_n),$$

*where*

$$b_i = \begin{cases} ka_i + b & \text{if } i \in J, \\ a_i & \text{otherwise.} \end{cases}$$

*Then $f$ is a continuous function.*

*Proof.* Consider two vectors $\mathbf{A}, \mathbf{B} \in \mathbb{R}^n$:

$$\mathbf{A} = (a_1, a_2, \ldots, a_n), \quad \mathbf{B} = (b_1, b_2, \ldots, b_n).$$

The mapping $f$ applies the linear transformation $kx + b$ to the elements of $\mathbf{A}$ indexed by $J$ and leaves the other elements unchanged:

$$f(\mathbf{A}) = (c_1, c_2, \ldots, c_n),$$

where

$$c_i = \begin{cases} ka_i + b & \text{if } i \in J, \\ a_i & \text{otherwise.} \end{cases}$$

Similarly,

$$f(\mathbf{B}) = (d_1, d_2, \ldots, d_n),$$

where

$$d_i = \begin{cases} kb_i + b & \text{if } i \in J, \\ b_i & \text{otherwise.} \end{cases}$$

The distance between the images of $\mathbf{A}$ and $\mathbf{B}$ under $f$ is given by

$$\|f(\mathbf{A}) - f(\mathbf{B})\| = \sqrt{\sum_{i=1}^{n}(c_i - d_i)^2}.$$

By the definition of $f$, we have

$$c_i - d_i = \begin{cases} k(a_i - b_i) & \text{if } i \in J, \\ a_i - b_i & \text{otherwise.} \end{cases}$$

Therefore, the distance can be rewritten as

$$\|f(\mathbf{A}) - f(\mathbf{B})\| = \sqrt{\sum_{i \in J}(k(a_i - b_i))^2 + \sum_{i \notin J}(a_i - b_i)^2}.$$

This simplifies to

$$\|f(\mathbf{A}) - f(\mathbf{B})\| = \sqrt{k^2 \sum_{i \in J}(a_i - b_i)^2 + \sum_{i \notin J}(a_i - b_i)^2}.$$

Let $C = \max(1, |k|)$. Then

$$\|f(\mathbf{A}) - f(\mathbf{B})\| \leq C\sqrt{\sum_{i=1}^{n}(a_i - b_i)^2} = C\|\mathbf{A} - \mathbf{B}\|.$$

Therefore, for any $\epsilon > 0$, choose $\delta = \frac{\epsilon}{C}$. If $\|\mathbf{A} - \mathbf{B}\| < \delta$, then

$$\|f(\mathbf{A}) - f(\mathbf{B})\| < C\delta = \epsilon.$$

Hence, $f$ is continuous. $\qquad\square$

**Theorem 4** (Pad Operation Continuity). *Let $\mathbf{A} = (a_1, a_2, \ldots, a_n) \in \mathbb{R}^n$. Define a mapping $f : \mathbb{R}^n \to \mathbb{R}^n$ such that for a fixed subset $J \subseteq \{1, 2, \ldots, n\}$ and a fixed constant $C \in \mathbb{R}$, we have*

$$f(\mathbf{A}) = (b_1, b_2, \ldots, b_n),$$

*where*

$$b_i = \begin{cases} C & \textit{if } i \in J, \\ a_i & \textit{otherwise}. \end{cases}$$

*Then $f$ is a continuous function.*

*Proof.* Consider two vectors $\mathbf{A}, \mathbf{B} \in \mathbb{R}^n$:

$$\mathbf{A} = (a_1, a_2, \ldots, a_n), \quad \mathbf{B} = (b_1, b_2, \ldots, b_n).$$

The mapping $f$ replaces each element of $\mathbf{A}$ at the positions indexed by $J$ with the constant $C$, and leaves the other elements unchanged:

$$f(\mathbf{A}) = (c_1, c_2, \ldots, c_n),$$

where

$$c_i = \begin{cases} C & \text{if } i \in J, \\ a_i & \text{otherwise}. \end{cases}$$

Similarly,

$$f(\mathbf{B}) = (d_1, d_2, \ldots, d_n),$$

where

$$d_i = \begin{cases} C & \text{if } i \in J, \\ b_i & \text{otherwise}. \end{cases}$$

The distance between the images of $\mathbf{A}$ and $\mathbf{B}$ under $f$ is given by

$$\|f(\mathbf{A}) - f(\mathbf{B})\| = \sqrt{\sum_{i=1}^{n}(c_i - d_i)^2}.$$

By the definition of $f$, we have

$$c_i - d_i = \begin{cases} 0 & \text{if } i \in J, \\ a_i - b_i & \text{otherwise}. \end{cases}$$

Therefore, the distance can be rewritten as

$$\|f(\mathbf{A}) - f(\mathbf{B})\| = \sqrt{\sum_{i \notin J}(a_i - b_i)^2}.$$

Note that the sum is only over the indices not in $J$. This is because the elements in $J$ are replaced by the constant $C$, and thus their difference is zero.

Since

$$\|f(\mathbf{A}) - f(\mathbf{B})\| \leq \|\mathbf{A} - \mathbf{B}\|,$$

for any $\epsilon > 0$, choose $\delta = \epsilon$. If $\|\mathbf{A} - \mathbf{B}\| < \delta$, then

$$\|f(\mathbf{A}) - f(\mathbf{B})\| < \epsilon.$$

Therefore, $f$ is continuous. $\qquad\square$

**Theorem 5** (Swap Operation Continuity). *Let $\mathbf{A} = (a_1, a_2, \ldots, a_n) \in \mathbb{R}^n$. Define a mapping $f : \mathbb{R}^n \to \mathbb{R}^n$ such that for fixed disjoint subsets $I, J \subseteq \{1, 2, \ldots, n\}$ with $I \cap J = \emptyset$ and $|I| = |J|$, the elements of $\mathbf{A}$ indexed by $I$ are swapped with the elements indexed by $J$. Then $f$ is a continuous function.*

*Proof.* Let $\mathbf{A} = (a_1, a_2, \ldots, a_n) \in \mathbb{R}^n$, and let $I = \{i_1, i_2, \ldots, i_m\}$ and $J = \{j_1, j_2, \ldots, j_m\}$ be two disjoint subsets of indices with $|I| = |J| = m$. Define the mapping $f$ such that it swaps the elements of $\mathbf{A}$ indexed by $I$ and $J$. Specifically, for any $\mathbf{A}$, the mapping $f$ produces a vector $\mathbf{B} = f(\mathbf{A})$ given by

$$b_k = \begin{cases} a_{j_r} & \text{if } k = i_r \text{ for some } r = 1, 2, \ldots, m, \\ a_{i_r} & \text{if } k = j_r \text{ for some } r = 1, 2, \ldots, m, \\ a_k & \text{otherwise.} \end{cases}$$

Consider two vectors $\mathbf{A}, \mathbf{C} \in \mathbb{R}^n$:

$$\mathbf{A} = (a_1, a_2, \ldots, a_n), \quad \mathbf{C} = (c_1, c_2, \ldots, c_n).$$

Applying the mapping $f$ to both vectors, we obtain

$$f(\mathbf{A}) = (b_1, b_2, \ldots, b_n), \quad f(\mathbf{C}) = (d_1, d_2, \ldots, d_n),$$

where

$$b_k = \begin{cases} a_{j_r} & \text{if } k = i_r \text{ for some } r = 1, 2, \ldots, m, \\ a_{i_r} & \text{if } k = j_r \text{ for some } r = 1, 2, \ldots, m, \\ a_k & \text{otherwise,} \end{cases}$$

and similarly,

$$d_k = \begin{cases} c_{j_r} & \text{if } k = i_r \text{ for some } r = 1, 2, \ldots, m, \\ c_{i_r} & \text{if } k = j_r \text{ for some } r = 1, 2, \ldots, m, \\ c_k & \text{otherwise.} \end{cases}$$

The distance between $f(\mathbf{A})$ and $f(\mathbf{C})$ is given by

$$\|f(\mathbf{A}) - f(\mathbf{C})\| = \sqrt{\sum_{k=1}^{n} (b_k - d_k)^2}.$$

Since $f$ only swaps the elements indexed by $I$ and $J$, we have

$$b_k - d_k = \begin{cases} a_{j_r} - c_{j_r} & \text{if } k = i_r \text{ for some } r, \\ a_{i_r} - c_{i_r} & \text{if } k = j_r \text{ for some } r, \\ a_k - c_k & \text{otherwise.} \end{cases}$$

Therefore, the norm becomes

$$\|f(\mathbf{A}) - f(\mathbf{C})\| = \sqrt{\sum_{r=1}^{m} (a_{j_r} - c_{j_r})^2 + \sum_{r=1}^{m} (a_{i_r} - c_{i_r})^2 + \sum_{k \notin I \cup J} (a_k - c_k)^2}.$$

Rearranging the terms, we have

$$\|f(\mathbf{A}) - f(\mathbf{C})\| = \sqrt{\sum_{k=1}^{n} (a_k - c_k)^2} = \|\mathbf{A} - \mathbf{C}\|.$$

Therefore, for any $\epsilon > 0$, choose $\delta = \epsilon$. If $\|\mathbf{A} - \mathbf{C}\| < \delta$, then

$$\|f(\mathbf{A}) - f(\mathbf{C})\| = \|\mathbf{A} - \mathbf{C}\| < \epsilon.$$

Hence, $f$ is continuous. $\qquad\square$

## C.2   COMPARISON WITH OTHER DISTANCE METRIC

We aim to explore whether using different distance metrics to define neighbor facts would also influence the model's inductive reasoning. Therefore, we additionally introduce three other distance metrics: Euclidean distance $d_{euc}$, Manhattan distance $d_{man}$, and Minkowski distance $d_{min}$. Like Equation 6, we have:

$$d_{euc}(\mathbb{X}_i, \boldsymbol{x}_t) = \sqrt{\sum_{k=1}^{D}(x_{ik} - x_{tk})^2} \tag{11}$$

$$d_{man}(\mathbb{X}_i, \boldsymbol{x}_t) = \sum_{k=1}^{D}|x_{ik} - x_{tk}| \tag{12}$$

$$d_{min}(\mathbb{X}_i, \boldsymbol{x}_t) = \left(\sum_{k=1}^{D}|x_{ik} - x_{tk}|^p\right)^{\frac{1}{p}} \tag{13}$$

where we set $p = 3$. We can generate three distinct new neighborhoods $\mathcal{N}(\boldsymbol{x}_t, \epsilon)$ by incorporating these distances into Equation 7, thereby constructing three new kinds of OF. Therefore, we compare the model's performance on EI tasks when using only these different OFs, and the results are shown in Figure 8. From the figure, we can see that **our neighborhood construction outperforms those constructed using other distance metrics**. The EI performance of the other three OFs across different radii is similar to the baseline, indicating that removing neighbor facts constructed using these methods does not influence the model's inductive reasoning ability. In contrast, our constructed OF leads to a significant decline in accuracy, proving the validity of our neighborhood construction.

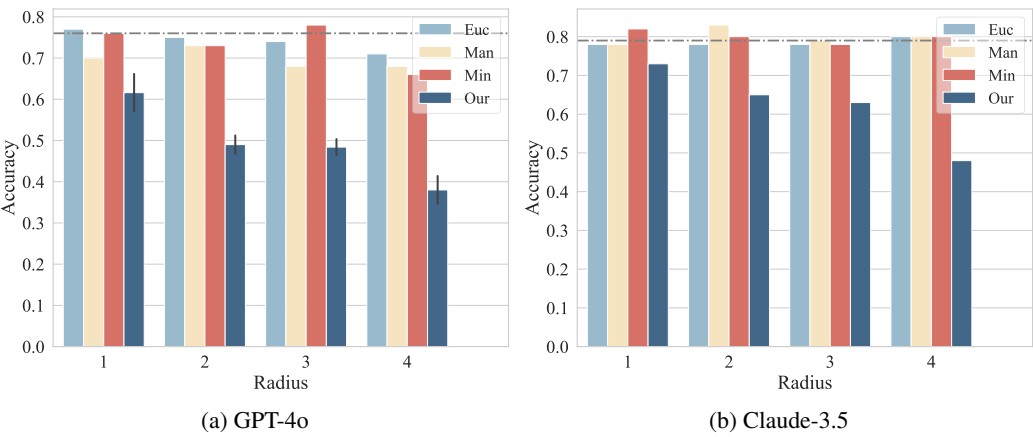

(a) GPT-4o                          (b) Claude-3.5

Figure 8: Performance comparison of the impact of different OFs. The dashed line represents the baseline accuracy using default fact sets. Euc represents Euclidean distance, Man represents Manhattan distance and Min represents Minkowski distance.

## C.3   MORE EXPERIMENTS ON OTHER MODELS

We repeat the experiments in § 4.3 on Llama2-13B, Claude-3.5 and Llama3-8B. The results are shown in Table 10, 11, 12. Besides, we also repeat the experiments in § 4.4 on Claude-3.5 and report the results in Figure 9. The results of all these additional experiments are consistent with those in the main text.

## C.4   SUPPLEMENTARY EXPERIMENT FOR MAIN EXPERIMENT

We observe that, in the experiment of § 4.2, though the performance of OF significantly decreases compared to the baseline, some models are still able to maintain around 40% accuracy, even with only distant observed facts. We infer that models are likely to conduct rule-based reasoning in these cases. Hence, we design an extra experiment for supplementary. In it, we prompt LLMs to induct rules and finish example inference tasks (i.e. ID in §3.2) on these cases in Table 13. From the table, we can observe that the model's deductive accuracy using the rule exceeds 70% when there are fewer neighbor facts in the context. This demonstrates that the model tends to rely more on rule-based induction if there is less neighbor-based matching.

| Type | N=3 | | | | N=5 | | | | N=8 | | | |
|------|-----|-----|-----|-----|-----|-----|-----|-----|-----|-----|-----|-----|
| | LT | RP | CG | ST | LT | RP | CG | ST | LT | RP | CG | ST |
| Baseline | 0.18 | 0.05 | 0.14 | 0.30 | 0.15 | 0.05 | 0.17 | 0.23 | 0.14 | 0.00 | 0.17 | 0.31 |
| IF Only | 0.43 | 0.14 | 0.35 | 0.34 | 0.48 | 0.03 | 0.49 | 0.36 | 0.46 | 0.02 | 0.52 | 0.36 |
| CF Only | 0.17 | 0.06 | 0.14 | 0.28 | 0.15 | 0.04 | 0.22 | 0.25 | 0.14 | 0.00 | 0.17 | 0.31 |
| OF Only | 0.16 | 0.04 | 0.13 | 0.27 | 0.09 | 0.02 | 0.09 | 0.26 | 0.10 | 0.01 | 0.10 | 0.22 |

Table 10: Performance of different fact types under various settings on Llama2-13B ($D = 5$).

| Type | N=3 | | | | N=5 | | | | N=8 | | | |
|------|-----|-----|-----|-----|-----|-----|-----|-----|-----|-----|-----|-----|
| | LT | RP | CG | ST | LT | RP | CG | ST | LT | RP | CG | ST |
| Baseline | 0.49 | 0.23 | 0.49 | 0.38 | 0.68 | 0.39 | 0.80 | 0.53 | 0.78 | 0.51 | 0.81 | 0.56 |
| IF Only | 0.76 | 0.42 | 0.84 | 0.61 | 0.90 | 0.46 | 0.87 | 0.61 | 0.89 | 0.62 | 0.93 | 0.76 |
| CF Only | 0.54 | 0.23 | 0.59 | 0.36 | 0.67 | 0.36 | 0.81 | 0.56 | 0.76 | 0.42 | 0.85 | 0.53 |
| OF Only | 0.50 | 0.24 | 0.49 | 0.45 | 0.60 | 0.30 | 0.71 | 0.39 | 0.66 | 0.26 | 0.77 | 0.49 |

Table 11: Performance of different fact types under various settings on Claude-3.5 ($D = 5$).

| Type | N=3 | | | | N=5 | | | | N=8 | | | |
|------|-----|-----|-----|-----|-----|-----|-----|-----|-----|-----|-----|-----|
| | LT | RP | CG | ST | LT | RP | CG | ST | LT | RP | CG | ST |
| Baseline | 0.19 | 0.12 | 0.26 | 0.29 | 0.27 | 0.17 | 0.28 | 0.32 | 0.24 | 0.26 | 0.37 | 0.20 |
| IF Only | 0.55 | 0.22 | 0.55 | 0.38 | 0.65 | 0.40 | 0.65 | 0.46 | 0.68 | 0.41 | 0.75 | 0.41 |
| CF Only | 0.21 | 0.07 | 0.29 | 0.29 | 0.27 | 0.14 | 0.31 | 0.31 | 0.27 | 0.22 | 0.39 | 0.19 |
| OF Only | 0.24 | 0.06 | 0.34 | 0.26 | 0.23 | 0.08 | 0.24 | 0.25 | 0.18 | 0.11 | 0.23 | 0.15 |

Table 12: Performance of different fact types under various settings on Llama3-8B ($D = 5$).

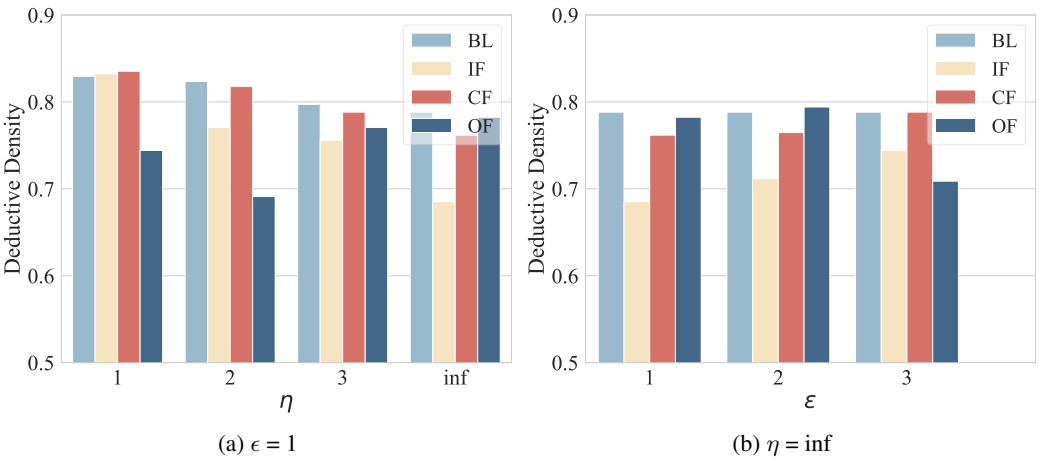

(a) $\epsilon = 1$

(b) $\eta = \inf$

Figure 9: Deductive Density of various fact types on Claude-3.5 under different test radius $\eta$ and neighborhood radius $\epsilon$ ($D = 5$, $N = 5$).

| Model | $\epsilon=1$ | $\epsilon=2$ | $\epsilon=3$ | Avg |
|-------|------|------|------|-----|
| GPT-4o | 0.74 | 0.72 | 0.76 | 0.74 |
| Claude-3.5 | 0.76 | 0.72 | 0.73 | 0.74 |

Table 13: Performance on the correct case of OF. We use the 0-shot setting and vary the radius $\epsilon$.

| Scenario | Prompt |
|---|---|
| **LT** | Please summarize the rules of the list transformation based on the given facts. |
| | Your reply should strictly follow the following format: |
| | Rule: [A, B, C] → [<<expression>>, <<expression>>, <<expression>>] |
| | Fact 1: Input: {INPUT}     Output: {OUTPUT} |
| | ... |
| | Fact n: Input: {INPUT}     Output: {OUTPUT} |
| | Please generate the rule of list transformation based on the former facts. |
| **RP** | Please summarize the rules of the {TASK_TYPE} based on the given facts. |
| | Your reply should strictly follow the following format: |
| | Rule: If there are A {OBJ1}, B {OBJ2}, C {OBJ3}. After the {TASK_TYPE}, |
| | there are <<expression>> {OBJ1}, <<expression>> {OBJ2}, <<expression>> {OBJ3}. |
| | Fact 1: Input: {INPUT}     Output: {OUTPUT} |
| | ... |
| | Fact n: Input: {INPUT}     Output: {OUTPUT} |
| | Please generate the rule of {TASK_TYPE} based on the former facts. |
| **CG** | Please summarize the rules of the function based on the given facts. |
| | Your reply should strictly follow the following format: |
| | Rule: |
| | def f(A, B, C): |
| |     A, B, C = <<expression>>, <<expression>>, <<expression>> |
| |     return A, B, C |
| | Fact 1: Input: {INPUT}     Output: {OUTPUT} |
| | ... |
| | Fact n: Input: {INPUT}     Output: {OUTPUT} |
| | Please generate the rule of function based on the former facts. |
| **ST** | Please summarize the rules of the string transformation based on the given facts. |
| | Your reply should strictly follow the following format: |
| | Rule: ABC → ... |
| | Fact 1: Input: {INPUT}     Output: {OUTPUT} |
| | ... |
| | Fact n: Input: {INPUT}     Output: {OUTPUT} |
| | Please generate the rule of string transformation based on the former facts. |

Table 14: Prompts for rule induction tasks ($D = 3$).

| Scenario | Prompt |
|---|---|
| LT | Please answer the question based on rules of the list transformation in the given facts.

Your reply should strictly follow the following format:

Answer: [<<expression>>, <<expression>>, <<expression>>]

Fact 1: Input: {INPUT}    Output: {OUTPUT}

...

Fact n: Input: {INPUT}    Output: {OUTPUT}

Question: Input: {TEST_INPUT} |
| RP | Please answer the question based on rules of the {TASK_TYPE} in the given facts.

Your reply should strictly follow the following format:

Answer: <<expression>> {OBJ1}, <<expression>> {OBJ2}, <<expression>> {OBJ3}.

Fact 1: Input: {INPUT}    Output: {OUTPUT}

...

Fact n: Input: {INPUT}    Output: {OUTPUT}

Question: Input: {TEST_INPUT} |
| CG | Please answer the question based on rules of the functioon in the given facts.

Your reply should strictly follow the following format:

Answer: <<expression>>, <<expression>>, <<expression>>

Fact 1: Input: {INPUT}    Output: {OUTPUT}

...

Fact n: Input: {INPUT}    Output: {OUTPUT}

Question: Input: {TEST_INPUT} |
| ST | Please answer the question based on rules of the string transformation in the given facts.

Your reply should strictly follow the following format:

Answer: ...

Fact 1: Input: {INPUT}    Output: {OUTPUT}

...

Fact n: Input: {INPUT}    Output: {OUTPUT}

Question: Input: {TEST_INPUT} |

Table 15: Prompts for example inference tasks ($D = 3$).

| Scenario | Template | Objects |
|---|---|---|
| **Trade** | {NAME} went to the market to trade items based on the rule. 
 He originally had {obj_expression} 
 After the trade, he had {obj_expression} | chairs, tables, pens ... |
| **Diet** | {NAME} adjusted his diet plan according to the expert's advice. 
 He originally planned to take in {obj_expression} 
 After the adjustment, he had {obj_expression} | apples, bananas, oranges ... |
| **Magic** | {NAME} was performing a card magic trick. 
 Initially, he had {obj_expression} 
 After performing the magic, he ended up with {obj_expression} | Spade 5s, Jokers, Hearts 6s ... |
| **Invest** | {NAME} adjusted the investment amount of each asset based on criteria. 
 Initially, he invested {obj_expression} 
 After the adjustment, he invested {obj_expression} | stocks, bonds, funds ... |
| **Course** | {NAME} adjusted the students' courses according to certain rules. 
 Initially, the weekly course schedule was: {obj_expression} 
 After the adjustment, the weekly course schedule was: {obj_expression} | math, science, history ... |

Table 16: Prompts for real-world problems construction.

