# OpenReview forum: "MIRAGE: Evaluating and Explaining Inductive Reasoning Process in Language Models"
_ICLR.cc/2025/Conference — ICLR 2025 Poster_

### Official Review · Reviewer_cTBS · 2024-10-29

**Soundness:** 2
**Presentation:** 3
**Contribution:** 2
**Rating:** 5
**Confidence:** 2

**Summary:**

Authors created a synthetic dataset MIRAGE to evaluate the reasoning ability of LLMs, with two conclusions: (1) LLMs are not good at deductive reasoning, (2) LLMs are good at performing inference based on neighborhood information, thus, they are "neighbor-based inductive reasoner".

Authors try to distinguish the ability of deductive reasoning from the ability of inductive reasoning. The missing experiment(s) is/are to distinguish the ability of reasoning from the ability of pattern matching. For example, it is necessary to show the so-called "neighbor-based inductive reasoner" is the ability of reasoning, instead of the ability of pattern-matching.

**Strengths:**

A new synthetic dataset is created and contributed to the reasoning evaluation of LLMs.
Authors conclude that LLMs are poor in rule-based reasoning, and good at neighbor-based reasoning.

**Weaknesses:**

Authors coined the nice term "neighbor-based reasoning". But, is it reasoning or just pattern-matching? I am afraid there is no term of "neighbor-based inductive reasoning" in the literature of logic or psychology.

It is not surprising that LLMs are good at "predicting" based on neighborhood context, as this is one of the important methods to train them.

**Questions:**

1. why do you only mention inductive reasoning in the title? you also report experiments on deductive reasoning of LLMs.

2. do you assume any causal relations between the input and the output of rules, or just associative relations? If only associative relations, how do you distinguish deductive from inductive reasoning?

---

> ### Author Response · Authors · 2024-11-13
> **Reply to Reviewer cTBS**
>
> Thanks for your careful and insightful reviews.
>
> + Question 1：Is “Neighborhood-Based Reasoning” reasoning or pattern matching?
>
> + Reply 1: We apologize for not making this point clear in the paper. In fact, the experiment in **Section 4.4** was conducted precisely to address this question. As you can see, in **Figure 7b and 7d**, we test the model's deductive performance on arbitrary test samples, and the results show that the model maintains a deductive density of **over 70%**  under the IF setting. This indicates that even when the patterns of the test samples differ significantly from those of the observed samples, the model can still perform reasoning through neighbor-based reasoning. **If this is a type of pattern matching, the model would only maintain high performance on nearby test samples, rather than sustaining such high performance globally**. Therefore, we believe that neighbor-based reasoning is **a form of reasoning rather than pattern matching**. Thanks for your suggestions, we will further emphasize this term in the revised version of our paper.
>
>
>
> + Question 2: It is not surprising that LLMs are good at "predicting" based on neighborhood context.
>
> + Reply 2: In fact, the neighborhood context in the next-word prediction task has, in essence, some fundamental differences from the neighboring facts identified in our work. For the former,  this type of "neighborhood" refers to **words that frequently co-occur in human language corpora**. However, for the latter, "neighborhood" refers to **the proximity between feature vectors in the vector space (see Section 4.1)**. This proximity in the mathematical space is rarely explicitly represented in the text. Therefore, it cannot be naturally assumed that training on the next-word prediction task will necessarily improve the model's ability for this type of neighbor-based reasoning.
>
>
>
> + Question 3: Why do you only mention inductive reasoning in the title?
>
> + Reply 3: We apologize that we don’t illustrate this term clearly. Actually, as we mentioned in **lines 33-36 of the introduction**, most related works incorporate both the **rule induction (inductive)** and **rule application (deductive)** phases into the inductive reasoning process [1,2,3]. Following them,  we consider the entire inductive reasoning process as a type of reasoning, with inductive and deductive being its two components. As you can see, **several representative related works also use "inductive reasoning" alone in their titles [1,2,3]**.
>
>
>
> + Question 4: The relations between the input and the output of rules? How do you distinguish deductive from inductive reasoning?
>
> + Reply 4: From my understanding, you may be concerned that rule induction might be an associative byproduct of the model's inductive reasoning process rather than a necessary phase.
>
>   + In fact, the experiments in **Section 3** are conducted to investigate the relationship. In prior work, from the perspective of human reasoning, rule induction and rule application are clearly divided into two separate phases [1,2,3]. Therefore, the input and the output of rules are viewed as having a causal relationship. However, our experiments challenge this viewpoint, suggesting that rule induction may not be necessary—that is, **it might simply be associative without implying causation**.
>   + We apologize that we don’t illustrate this term clearly. Actually, our work aims to investigate the role of rule induction in the overall inductive reasoning process,  **without focusing on specific types of relationships**. We can measure the accuracy of induction through a rule-based question-answering task, thereby **assessing the correlation** between rule induction and the final performance. Measuring this correlation **does not require a strict distinction** between inductive and deductive reasoning. In this paper, we simply distinguish between inductive and deductive **based on the form of the test questions rather than the formal definitions in logic**.
>
>   Thanks for your suggestions, we will make further discussions on this term in the revised version of our paper. If I have misunderstood your question in any way, please feel free to point it out.
>
>
>
> [1] Wang, Ruocheng, et al. "Hypothesis Search: Inductive Reasoning with Language Models." *The Twelfth International Conference on Learning Representations*.
>
> [2] Qiu, Linlu, et al. "Phenomenal Yet Puzzling: Testing Inductive Reasoning Capabilities of Language Models with Hypothesis Refinement." *The Twelfth International Conference on Learning Representations*.
>
> [3] Bowen, Chen, Rune Sætre, and Yusuke Miyao. "A Comprehensive Evaluation of Inductive Reasoning Capabilities and Problem Solving in Large Language Models." *Findings of the Association for Computational Linguistics: EACL 2024*. 2024.

---

> > ### Comment · Reviewer_cTBS · 2024-11-15
> >
> > Thanks for the reply. I am a bit confused. "Neighborhood-Based Reasoning" can be used both for deductive reasoning and inductive reasoning, depending on which premises we use. If we have the region as the premise that follows any point in the region having a feature, it will be deductive reasoning. If we have a point as the premise to infer points nearby (defining a region) having the same feature, it will be inductive reasoning. Your experiment and your explanation are a good mixture of two.

---

> > > ### Author Response · Authors · 2024-11-15
> > > **Reply to Reviewer cTBS**
> > >
> > > Thanks for your response and we apologize once again for any confusion caused by our unclear writing. In our work, the inductive task and deductive task are two perspectives we use to measure the model's inductive reasoning capability. **The neighbor-based reasoning paradigm is a key factor affecting this capability, and therefore it influences the performance on both tasks.** Our experiments in **Sections 4.2 and 4.3** mainly demonstrate that this pattern has a significant impact on the model's performance on local examples (i.e., **deductive**). The experiments in **Section 4.4** primarily discuss the global influence of this pattern (i.e., **inductive**), showing that it can be effective within a certain scope. Therefore, we find that this paradigm plays a role in both tasks, further demonstrating its importance. For the premises, we maintain consistency across different tasks. It is challenging for the model to perform this kind of reasoning using just a single point as a premise (similar to humans), so we **fix it as a set of points under a specified distribution** (i.e., BL, IF, CF, OF) like previous works. The difference between the two tasks lies in the scope of testing rather than the premises.

---

> > > > ### Author Response · Authors · 2024-11-23
> > > > **Reply to Reviewer cTBS**
> > > >
> > > > Dear Reviewer
> > > >
> > > > We sincerely appreciate the time and effort you have dedicated to reviewing our work. In response to your valuable feedback, we have provided detailed clarifications to the questions raised and included a revised version of the paper. As we are nearing the end of the discussion period, we would love to hear your thoughts on our response, including whether it sufficiently addresses your concerns. If our revisions and discussions indicate potential for a score adjustment, we would be deeply grateful for your consideration. We remain committed to incorporating all your suggestions to further enhance the quality of our manuscript. We hope we have addressed all your concerns and look forward to your further comments and discussions.

---

> > > > > ### Comment · Reviewer_cTBS · 2024-11-24
> > > > >
> > > > > I read your comment to Reviewer DD2X: "It is worth noting that our deductive task differs from traditional logic as it does not take rules as input." This difference from well-known terms in traditional logic is really misleading. You should not use this way.

---

> > > > > > ### Author Response · Authors · 2024-11-24
> > > > > > **Reply to Reviewer cTBS**
> > > > > >
> > > > > > Dear Reviewer cTBS
> > > > > >
> > > > > > Thank you for your feedback. We sincerely apologize for any misunderstanding caused by this naming convention. However, using observed facts as input (without explicit rules) to directly test on new facts is a widely adopted approach in NLP research on inductive reasoning. In our case, since we introduced an additional task that directly tests the rules themselves, we named the former as the deductive task to **distinguish between the two**. We apologize once again for the confusion this naming might have caused. Therefore, in the revised version of the paper, we will rename the two tasks as the "**rule induction task**" and the "**example inference task**" to avoid such misunderstandings. If you have any further suggestions regarding this adjustment, please feel free to share them with us at any time.

---

> > > > > > > ### Author Response · Authors · 2024-11-25
> > > > > > > **Reply to Reviewer cTBS**
> > > > > > >
> > > > > > > We sincerely appreciate your feedback. In response, we have provided detailed clarifications to the questions raised and an updated version of our manuscript. Specifically, to avoid confusion caused by the naming and its potential overlap with definitions in traditional logic, we have renamed the original inductive task and deductive task in this paper as the **rule induction task** and the **example inference task**. We have also clarified their specific definitions and roles in the overall inductive reasoning process (**see lines 186-196**). It is important to emphasize that these naming adjustments **do not affect any of the experimental conclusions or contributions of this work**. Thank you again for your valuable suggestions. We are eager to know whether our revisions have addressed your concerns, and we warmly encourage you to share your thoughts with us.

---

> > > > > > ### Author Response · Authors · 2024-11-29
> > > > > > **Reply to Reviewer cTBS**
> > > > > >
> > > > > > Thank you for your response and for raising the score. If you have any further questions or concerns, please feel free to contact us at any time.

---

### Official Review · Reviewer_ctTK · 2024-11-03

**Soundness:** 3
**Presentation:** 3
**Contribution:** 3
**Rating:** 8
**Confidence:** 5

**Summary:**

This paper investigates the inductive reasoning abilities of LLMs, introducing Mirage, a synthetic dataset designed to comprehensively evaluate LLMs' inductive and deductive reasoning from various perspectives. Using Mirage, the authors demonstrate that LLMs are weak at rule-based reasoning but perform well with neighbor-based reasoning. These findings suggest that LLMs' deductive reasoning can be enhanced by leveraging similar examples to support a robust inductive reasoning process.

**Strengths:**

Strengths:
- Present a synthetic data generation framework to evaluate inductive and deductive reasoning processes. The authors provide clear steps detailing the data generation process, specifically outlining the rule generation and question generation procedures.
- Various advanced prompting techniques, including SR and HR, along with the latest LLMs, are tested in the experiments.
- The effective scope analysis for neighbor reasoning supports the claim that LLMs are effective neighbor-based reasoners.

**Weaknesses:**

Weaknesses:
- In the section on neighbor reasoning, the authors use the term "Feature Space" to denote the space of observed facts and test samples. However, it’s unclear where the features in this space originate.
- The data setting may lack sufficient complexity to effectively demonstrate inductive reasoning capabilities in real-life scenarios.

**Questions:**

Questions:
- Could the authors offer additional insights or empirical evidence on the model's performance with real-world datasets or in less controlled environments?
- Additionally, could the authors also check if fine-tuning over the synthetic data helps the inductive reasoning of LLMs?

---

> ### Author Response · Authors · 2024-11-14
> **Reply to Reviewer ctTK**
>
> Thanks for your careful and insightful reviews.
>
> + Question 1: It’s unclear where the features in the feature space originate.
> + Reply 1: We apologize for not making this point more clear in the paper. Actually, we mention the feature in **lines 368-370**, which refers to the input vectors of different facts. Thanks for your suggestions, we will further emphasize this term in the revised version of our paper.
>
>
>
> + Question 2: The data setting may lack sufficient complexity to effectively demonstrate inductive reasoning capabilities in real-life scenarios.
> + Reply 2: As we mention in **lines 181-184**, in Mirage, we design the scenario of **real-world problem (RP)** to evaluate the inductive reasoning capabilities in real-life scenarios. Based on **Table 1**, in this task, all of the advanced LLMs achieve an accuracy of **less than 0.25** (D=8). This indicates that the task is already very challenging for models, further demonstrating that the task complexity is sufficient for current models (even though they are relatively easy for humans).
>
>
>
> + Question 3: Additional insights or empirical evidence on the model's performance with real-world datasets or in less controlled environments.
> + Reply 3: We show the experimental results of real-world problems (RP) in **Table 1 and 2**. Based on them, we demonstrate that models show poor performance on them. We infer that this is due to the presence of a large amount of irrelevant natural language text in real-world problems, which makes it significantly challenging for LLMs to extract mathematical patterns (e.g. rules, facts) from the context.
>
>
>
> + Question 4: Does fine-tuning over the synthetic data help the inductive reasoning of LLMs?
>
> + Reply 4: Thanks for your suggestions. Here, we fine-tune the Llama3-8b-chat model using LoRA on 8,000 training examples and evaluate its performance on the test set. We report the performance of the list transformation task with D=5, N=5 as follows:
>
>   | Task      | ICL  | Fine-tune    |
>   | --------- | ---- | ------------ |
>   | Inductive | 0.31 | 0.89 (+0.58) |
>   | Deductive | 0.27 | 0.82 (+0.55) |
>
>   The results demonstrate that fine-tuning the model on synthetic data can significantly improve its inductive reasoning capabilities. We will report the details and complete results of this experiment in the revised version of the paper once all experiments are completed.

---

> > ### Comment · Reviewer_ctTK · 2024-11-14
> >
> > Thank you to the authors for providing clarification on my previous questions. Regarding Question 4, I would suggest fine-tuning the model on a single type of synthetic data, such as list transformation, and then evaluating its performance on other scenarios, such as RP and CG. This approach could provide clearer insights into the model's generalization ability across diverse data types. I maintain my current score, and improve my confidence.

---

> > > ### Author Response · Authors · 2024-11-14
> > > **Reply to Reviewer ctTK**
> > >
> > > Thanks for your response and the raise of confidence. We will add these experiments in the revised version of our paper.

---

### Official Review · Reviewer_DD2X · 2024-11-03

**Soundness:** 3
**Presentation:** 4
**Contribution:** 3
**Rating:** 6
**Confidence:** 4

**Summary:**

This paper presents a new dataset for evaluating the induction and deduction ability of LLMs. The authors provide a very detailed explanation of the generation method of the dataset and make extensive experiments to evaluate current state-of-the-art models. The authors also propose two hypotheses about LLMs' reasoning behaviours, and the experiments are designed to verify these hypotheses. Detailed explanations and discussion are provided, and the supplementary material also gives good support to the conclusion.

**Strengths:**

- The paper covered a broad range of related works and provided another valuable benchmark for evaluating LLMs' reasoning power. The tasks has been formulated in four representations to ensure the diversity.
- The authors make two interesting hypotheses on LLM's behaviour, and carefully designed several experiments to verify them.
- In order to evaluate LLMs' behaviours, the authors designed several metrics specifically, most of which make sense to me. Detailed discussions are provided, and all the charts and tables are explained well.

**Weaknesses:**

- First of all, all the experiments are behavioural studies and empirical evaluations, and there is no mathematical evidence about LLM's induction/deduction power. IMHO, if there is no mathematical proof, then claims such as "we prove that LLMs are poor in inductive reasoning" would be too strong.
- It is fine to evaluate the models' induction ability by using the designed tasks. However, I wouldn't call the other task as "deduction" since the rules are not provided. From the definition from logic, a deductive inference needs both input facts and the logic rules for deducing the outputs.

**Questions:**

Please see above.

---

> ### Author Response · Authors · 2024-11-13
> **Reply to Reviewer DD2X**
>
> Thanks for your careful and insightful reviews.
>
> + Question 1: Claims such as "we prove that LLMs are poor in inductive reasoning" would be too strong.
> + Reply 1: We apologize for the lack of clarity in our expression here. Thanks for your suggestion, we will modify these claims in our revised version. For example, "**We demonstrate that LLM is a relatively poor rule-based inductive reasoner compared to direct deduction**".
>
>
>
> + Question 2: A deductive inference needs both input facts and the logic rules for deducing the outputs.
> + Reply 2: We apologize for not making this point clear in the paper. Here, we distinguish between inductive and deductive **based on the form of the test questions rather than the formal definitions in logic**. In our paper, we aim to investigate whether the model must undergo rule induction to complete deductive tasks. Therefore, we design this task form without including the rule as input,  **allowing us to directly evaluate the model's performance on the deductive test examples**. Besides, we also **implement rule-based inference in the ID method described in Section 3.2**, and the experiments conducted with this method also support our conclusion. Thanks for the suggestion, we will further clarify this inconsistency in the revised version of our paper.

---

> > ### Author Response · Authors · 2024-11-23
> > **Reply to Reviewer DD2X**
> >
> > Dear Reviewer
> >
> > We sincerely appreciate the time and effort you have dedicated to reviewing our work. In response to your valuable feedback, we have provided detailed clarifications to the questions raised and included a revised version of the paper. As we are nearing the end of the discussion period, we would love to hear your thoughts on our response, including whether it sufficiently addresses your concerns. If our revisions and discussions indicate potential for a score adjustment, we would be deeply grateful for your consideration. We remain committed to incorporating all your suggestions to further enhance the quality of our manuscript. We hope we have addressed all your concerns and look forward to your further comments and discussions.

---

> > ### Comment · Reviewer_DD2X · 2024-11-24
> >
> > > We will modify these claims in our revised version. For example, "We demonstrate that LLM is a relatively poor rule-based inductive reasoner compared to direct deduction".
> >
> > Great. The updated description looks much more accurate.
> >
> > > Here, we distinguish between inductive and deductive based on the form of the test questions rather than the formal definitions in logic. In our paper, we aim to investigate whether the model must undergo rule induction to complete deductive tasks. Therefore, we design this task form without including the rule as input, allowing us to directly evaluate the model's performance on the deductive test examples. Besides, we also implement rule-based inference in the ID method described in Section 3.2, and the experiments conducted with this method also support our conclusion. Thanks for the suggestion, we will further clarify this inconsistency in the revised version of our paper.
> >
> > Thank you for the explanation, but I still think it is difficult to specify tasks with implicit induction just from the behaviour of ML models/humans. Would it be possible to provide a clear definition of this in the paper? I think even a informal one would be good.

---

> > > ### Author Response · Authors · 2024-11-29
> > > **Reply to Reviewer DD2X**
> > >
> > > Dear Reviewer
> > >
> > > Thank you for your invaluable suggestions, which have greatly contributed to improving our paper. We would be truly grateful for your feedback on our revisions. Have our responses addressed your concerns, or are there additional issues you would like to discuss? As we are nearing the end of the discussion period, we hope to have the opportunity to further engage with you on these matters. We sincerely look forward to hearing your thoughts and greatly value your input!

---

> ### Author Response · Authors · 2024-11-24
> **Reply to Reviewer DD2X**
>
> Thank you for your response and valuable suggestions. We will incorporate the following content into the revised version of the paper to better clarify the definition of our tasks:
>
> For humans, although the process of reasoning tends to be implicit, according to the traditional definition in logic, it can be divided into two stages: induction and deduction. Here, we focus on the objectives of these two stages: deriving the correct rules through induction and making inferences on new instances using deduction. Therefore, we define the following two tasks in Mirage:
>
> + **Definition 1 (Rule Induction Task):** Given an observed fact set, this task evaluates whether the model can successfully **infer the underlying transformation rules** behind these facts
> + **Definition 2 (Example Inference Task):** Given an observed fact set, this task evaluates whether the model can successfully **perform inference on unseen instances** that follow the same rules.
>
> It is worth noting that our second task differs from traditional deductive tasks as it does not take rules as input. Its purpose is to evaluate the model's final performance on new instances, without requiring the model to explicitly undergo a rule induction process. Therefore, we can assess the model's reliance on rules in its reasoning process by comparing its performance on these two tasks.
>
> If you have any remaining questions or concerns about our response, we would be happy to engage in further discussion with you.

---

> > ### Author Response · Authors · 2024-11-25
> > **Reply to Reviewer DD2X**
> >
> > We sincerely appreciate your feedback. In response, we have provided detailed clarifications to the questions raised and an updated version of our manuscript. Specifically, to avoid confusion caused by the naming and its potential overlap with definitions in traditional logic, we have renamed the original inductive task and deductive task in this paper as the rule induction task and the example inference task. We have also **clarified their specific definitions and roles in the overall inductive reasoning process** (**see lines 186-196**). Thank you again for your valuable suggestions. We are eager to know whether our revisions have addressed your concerns, and we warmly encourage you to share your thoughts with us.

---

### Official Review · Reviewer_6pHZ · 2024-11-04

**Soundness:** 2
**Presentation:** 3
**Contribution:** 3
**Rating:** 5
**Confidence:** 3

**Summary:**

This paper evaluates and analyzes LLM's ability for inductive reasoning. Compared to previous works in inductive reasoning, it adopts a new setting, which is to compare the performance of LLMs under two settings: (1) specific --> general --> specific, and (2) specific --> specific.
It is interesting to see that the performance of LLMs under the two settings is comparable. The result indicates that setting (2) will have a better performance, even though traditionally we think a general rule is important to make further inferences to unseen specific examples.

As for results, the main findings of this paper are that
(1) setting (2) outperforms setting (1);
(2) many techniques such as self-consistency can't improve performance in the proposed task; and
(3) in-context demonstrations that are closer to the input can help more in terms of the accuracy of output.

**Strengths:**

1. The main finding is insightful, that setting (2) outperforms setting (1), which can bring in more discussions on how LLMs perform inductive reasoning.
2. The analysis in terms of neighbor-based reasoning, especially the investigation into the scope is also interesting. However, it might not be surprising, since the similarity of in-context demonstrations has been discussed extensively in previous papers.

**Weaknesses:**

1. The writing is not clear. Specifically,
a). what does the "substitution" in line 414 and line "443" mean?
b). after finding IF, CF, and OF, how are they used?
c). what is the metric in Table 1?

2. The claim "We prove that LLM is a poor rule-based inductive reasoner" is too strong and unacceptable. What is the definition of "poor"? And have the authors proved it mathematically?

3. "We prove that LLM is a neighbor-based inductive reasoner.": Have the authors proved it mathematically? When LLM is a neighbor-based inductive reasoner?

4. Although the performance that setting (2) outperforms setting (1) is provided. There might lack of a more in-depth analysis of when LLMs use neighbor-based reasoning, and when LLMs use the general pattern thinking? Or is it that LLMs can purely do neighbor-based reasoning, and never use the general pattern thinking at all? It seems that although each experiment is interesting, but I don't know what is the takeaway knowledge from this paper.

**Questions:**

See above

---

> ### Author Response · Authors · 2024-11-13
> **Reply to Reviewer 6pHZ**
>
> Thanks for your careful and insightful reviews.
>
> + Question 1：The writing is not clear.
>
> + Reply 1：We apologize that we don’t illustrate these terms clearly. Here we provide additional details regarding them.
>
>   + a) As we mention in lines 414-415, "**make the fact set X contain only one type of the fact**". For each fact in the fact set, if it is not of the specified type, we remove it and regenerate a new fact until all facts in the set are of the specified type (i.e., IF, CF, or OF).
>   + b) As we mention in a), after finding IF, CF and OF, we replace facts that do not meet the definition. Therefore, we ensure that the entire fact set contains only one type of fact, which **allows the model to observe and perform the corresponding deductive task under different settings**.
>   + c) As  we emphasize in **lines 194-195**, the fundamental metric for our entire dataset is **accuracy**.
>
>   Thanks for your suggestions, we will further emphasize these terms in the revised version of our paper.
>
>
>
> + Question 2: The claim "We prove that LLM is a poor rule-based inductive reasoner" is too strong and unacceptable. What is the definition of "poor"? And have the authors proved it mathematically?
>
> + Reply 2: We apologize for the lack of clarity in our expression here. Here, "poor" refers to **the model's low reliance on rules during the inductive reasoning process**. This does not refer to absolute performance, but rather indicates that the model's performance on inductive tasks is **relatively poor** compared to its performance on deductive tasks. For the proof, we believe that the poor reasoning performance of the model can only be demonstrated experimentally, not proven mathematically. Therefore, **through the experiments in Section 3, we provide empirical evidence to support this claim**. Thanks for your suggestion, we will modify this claim to "**We demonstrate that LLM is a relatively poor rule-based inductive reasoner compared to direct deduction**" in our revised version.
>
>
>
> + Question 3: Have the authors proved the neighbor-based inductive reasoning mathematically? When LLM is a neighbor-based inductive reasoner?
>
> + Reply 3: We apologize for the lack of clarity in our expression here. Like we mention in Reply 2, the reasoning performance can not be proven mathematically. Instead, we **provide empirical evidence to support this claim through the experiments in Section 4**. Thanks for your suggestion, we will modify this claim to "**We demonstrate that LLM is a neighbor-based inductive reasoner**" in our revised version. For the second question, we provide a detailed response in Reply 4.
>
>
>
> + Question 4: Lack of a more in-depth analysis of when LLMs use neighbor-based reasoning, and when LLMs use the general pattern thinking?
>
> + Reply 4: In fact, we have conducted a preliminary exploration of this question in **Appendix C.4.** From the experimental results in Table 12, we can conclude that the model mainly uses the general pattern thinking **when there are few neighbor facts in the context** (e.g. OF). Otherwise, they rely more on neighbor-based reasoning. However, in some cases, the two paradigms may still be coupled together, and therefore, we cannot categorically exclude the influence of either paradigm on the reasoning process.

---

> > ### Author Response · Authors · 2024-11-23
> > **Reply to Reviewer 6pHZ**
> >
> > Dear Reviewer
> >
> > We sincerely appreciate the time and effort you have dedicated to reviewing our work. In response to your valuable feedback, we have provided detailed clarifications to the questions raised and included a revised version of the paper. As we are nearing the end of the discussion period, we would love to hear your thoughts on our response, including whether it sufficiently addresses your concerns. If our revisions and discussions indicate potential for a score adjustment, we would be deeply grateful for your consideration. We remain committed to incorporating all your suggestions to further enhance the quality of our manuscript. We hope we have addressed all your concerns and look forward to your further comments and discussions.

---

> ### Author Response · Authors · 2024-11-26
> **Reply to Reviewer 6pHZ**
>
> Dear Reviewer
>
> Thank you for your invaluable suggestions, which have greatly contributed to improving our paper. We would be truly grateful for your feedback on our revisions. Have our responses addressed your concerns, or are there additional issues you would like to discuss? With the discussion period extended, we hope to have the opportunity to further engage with you on these matters. We sincerely look forward to hearing your thoughts and greatly value your input!

---

> ### Author Response · Authors · 2024-11-30
> **Gentle Reminder Regarding Rebuttal Feedback**
>
> Dear Reviewer 6pHZ,
>
> I hope this message finds you well.
>
> First and foremost, please allow me to extend our sincere apologies for the inconvenience caused by this follow-up correspondence. We are fully aware of the demands of your time and appreciate the dedication and effort you have already shown during the review process of our manuscript.
>
> **As we are nearing the end of the rebuttal phase, we are keen to ensure that all concerns have been adequately addressed**. We have made every attempt to respond to each of your comments with the utmost care and attention. Our goal is to ensure that our manuscript is as strong and as responsive to the reviewers' feedback as possible.
>
> We kindly request your confirmation on whether our responses have satisfactorily resolved your concerns. **As the deadline for the rebuttal is drawing near, we find ourselves in a position where we need to seek a status update on your review**. We are fully aware that the review process can be time-consuming, and we appreciate your efforts in this regard.
>
> Once again, we apologize for any disturbance and thank you for your understanding and support.

---

> ### Comment · Reviewer_6pHZ · 2024-12-02
>
> I appreciate the author's response.
>
> For Q3, "We demonstrate that LLM is a neighbor-based inductive reasoner", I guess some context should be needed? Do the authors imply that LLMs always conduct neighbor-based inductive reasoning for any questions?
>
> For Q4, I have checked Appendix C.4, but it seems more details can help it more clear. For example, "... exceeds 70% when there are fewer neighbor facts in the context. This demonstrates that the model tends to rely more on rule-based induction". What is the logic here? The authors in their rebuttal suggest to take a look at Table 12, but no context in the paper has referred to Table 12. It is unclear what message can be obtained from it. Can the authors give more explanations to Appendix C.4 to support the existing conclusions? A clear and persuading anslysis on the takeway message (e.g., on when LLMs use neighbor-based reasoning, and when LLMs use the general pattern thinking/rule-based reasoning?) could largely benefit the paper.
>
> I would also suggest the authors to take care of the conclusions. For example in the conclusion section, "we demonstrated that LLM is a poor rule-based reasoner, it does not need to rely on inductive rules when performing inductive reasoning." --- what does it mean to be a "poor rule-based reasoner"? Does it only mean LLM doesn't rely on rules to perform reasoning, or does it also mean LLM can't induce rules? Similar problems are very common throughout the paper. I would also suggest the authors to check each conclusion made in this paper on overclaim or ambiguity issues.
>
> I have increased the presentation score from 2 to 3. I will continue update the overall rating if
> (1) a clear and persuading anslysis on the takeway message can be provided, and
> (2) the overclaim and ambiguity issues on conclusions are properly solved in this paper.

---

> > ### Author Response · Authors · 2024-12-02
> > **Reply to Reviewer 6pHZ (part 1)**
> >
> > Thanks for your careful and insightful reviews. Once again, we sincerely apologize for the lack of clarity in our writing. Below are our responses to the questions you raised:
> >
> > ### **Takeaway Messages in Our Paper**
> >
> > 1. **For rule-based reasoning:**
> >
> >    + **Conclusion**
> >
> >      In inductive reasoning, the model demonstrates a relatively weak dependence on rules when performing example inference. Although the model possesses some rule induction capability, it can perform correct example inference without the need for successful rule induction.
> >
> >    + **Evidence & Analysis**
> >
> >      From Tables 1 and 3, we can see that the model's performance on the example inference task is significantly higher than that on the rule induction task. This demonstrates that successful example inference of a model does not require correct induction of the corresponding rules. This finding challenges the approach adopted in previous related work [1,2,3], which relied on generating correct rules before performing example inference, without recognizing the existence of such a shortcut.
> >
> > 2. **For neighbor-based reasoning:**
> >
> >    + **Conclusion**
> >
> >      When neighbor facts are present in the observed examples, the model leverages these examples directly to perform the example inference. The neighbor facts in the context have a great impact on the model's inductive reasoning ability. When there are more neighbor facts, the example inference performance of the model will be enhanced, and vice versa.
> >
> >    + **Evidence & Analysis**
> >
> >      In Table 4 and Figure 5, we can see that when neighbor facts are present in the model (i.e., under the IF settings), the model's inductive reasoning performance is significantly higher compared to when neighbor facts are absent (i.e., under the OF settings). This indicates that these neighbor facts have a substantial impact on the model's example inference performance. This provides an explanation for the previously identified shortcut: when neighbor facts are present, the model can perform example inference directly by leveraging these facts without relying on rules. This is a novel conclusion that has not been identified in previous related work. It highlights and explains the existence of a shortcut reasoning pattern in the model's inductive reasoning process.
> >
> >
> >
> > ### **Explanation of Experiments in Appendix C.4**
> >
> > + **Analysis**
> >
> >   This experiment primarily investigates the model's reasoning patterns in the absence of neighbor facts. We ensure the absence of neighbor facts in the context by setting OF (i.e. all facts are located at a considerable distance from the test example.).  We employed the ID method (i.e., the model needs to first induce correct rules and then perform inference) to evaluate the model's rule induction performance on all successful cases. The results showed that the success rate averaged over 70% (as shown in Table 13, previously Table 12).
> >
> > + **Conclusion**
> >
> >   This indicates that, in the absence of neighbor facts, the model tends to successfully induce rules when it achieves successful example inference. In such cases, the model demonstrates an ability to generalize global rules rather than relying on example-to-example shortcuts. This further supports our takeaway messages, which states that in the absence of neighbor facts, the model struggles to perform neighbor-based reasoning.

---

> > ### Author Response · Authors · 2024-12-02
> > **Reply to Reviewer 6pHZ (part 2)**
> >
> > ### **Revisions for Overclaim and Ambiguity Issues**
> >
> > + **Statement 1**
> >
> >   + **Original version**
> >
> >      *We demonstrate that LLM is a neighbor-based inductive reasoner.*
> >
> >   + **Revised version**
> >
> >      *We demonstrate that the model utilizes neighbor facts in the context to assist its inference on new examples. when neighbor facts are available in inductive reasoning.*
> >
> >   + **Explanation**
> >
> >     Here, the context refers to when neighbor facts are present in the observed examples.
> >
> > + **Statement 2**
> >
> >   + **Original version**
> >
> >     *We demonstrate that LLM is a poor rule-based reasoner, it does not need to rely on inductive rules when performing inductive reasoning.*
> >
> >   + **Revised version**
> >
> >     *We demonstrate that in inductive reasoning, the model's rule induction capability is significantly weaker than its example inference capability. When performing inference on new examples, the model does not necessarily rely on successful rule induction.*
> >
> >   + **Explanation**
> >
> >     Here, "poor" implies that the model's rule induction performance is poorer compared to its example inference performance and that the model's reliance on rules during inference is also relatively poor.
> >
> > We assure you that all overclaim and ambiguity issues will be addressed and improved in the revised version of the paper.
> >
> >
> >
> > If you have any remaining questions or concerns about our response, we would be happy to engage in further discussion with you. We sincerely look forward to hearing your thoughts and greatly value your input.
> >
> >
> >
> > [1] Wang, Ruocheng, et al. "Hypothesis Search: Inductive Reasoning with Language Models." *The Twelfth International Conference on Learning Representations*.
> >
> > [2] Qiu, Linlu, et al. "Phenomenal Yet Puzzling: Testing Inductive Reasoning Capabilities of Language Models with Hypothesis Refinement." *The Twelfth International Conference on Learning Representations*.
> >
> > [3] Bowen, Chen, Rune Sætre, and Yusuke Miyao. "A Comprehensive Evaluation of Inductive Reasoning Capabilities and Problem Solving in Large Language Models." *Findings of the Association for Computational Linguistics: EACL 2024*. 2024.

---

> > ### Author Response · Authors · 2024-12-03
> > **Reply to Reviewer 6pHZ**
> >
> > Dear Reviewer
> >
> > As the discussion period is nearing its end, we would like to know if our response has adequately addressed your concerns. If you have any remaining questions or doubts, please do not hesitate to provide your feedback. We will do our utmost to resolve any issues you may have. We would like to once again express our gratitude for your efforts and contributions to improving our paper.

---

### Author Response · Authors · 2024-11-18
**Response to Reviewers**

Dear Reviewers

We sincerely appreciate the time and effort you've dedicated to reviewing our work. In response to your valuable feedback, we have provided detailed clarifications to the questions raised and updated the version of our manuscript, mainly including:

+ **More rigorous expression:** Based on Reviewer 6pHZ's questions 2 and 3, as well as Reviewer DD2X's question 1, we have **revised the main contributions of the paper** to: "We find that LLM is a poor rule-based inductive reasoner" (see line 105) and "We demonstrate that LLM is a neighbor-based inductive reasoner" (see line 106).
+ **Clearer task categorization:** To address the confusion regarding inductive and deductive reasoning raised in Reviewer DD2X and Reviewer cTBS, we have added a footnote in line 44 of the Introduction to **clarify the task definition and distinguish it from traditional logic**.
+ **Clearer presentation:** We have **emphasized some of the original but unclear expressions in our manuscript**. For example, Reviewer 6pHZ's question 1 (refer to line 414 and Table 1) and Reviewer cTBS's question 1 (refer to lines 455-456).
+ **More Experiments**: Following the suggestion of Reviewer ctTK's question 4, we have **added experiments in Appendix B.3 to explore the impact of fine-tuning methods** on the model's inductive reasoning capabilities.

We sincerely apologize once again for the confusion caused by these presentation issues. At the same time, **we eagerly await your thoughts on our response**, particularly whether it adequately addresses your concerns. We remain fully committed to incorporating all of your suggestions into our revision process to further improve the quality of our manuscript.

---

### Meta-Review · Area_Chair_iCdN · 2024-12-19

**Metareview:**

The paper claims that LLMs perform better when asked to generalize from facts to neighbor facts (dubbed neighbor-based reasoning) than when asked to learn rules from the available facts, and thereafter use these rules (rule-based reasoning).
A dataset MIRAGE is built to investigate and experimentally support this claim. The dataset generation proceeds by defining transformation rules, synthesizing facts based on these rules, designing inductive and deductive questions on the facts, and designing tasks (list and string transformations, code generations).
The importance of the filtering step should be assessed too; can fact or rule redundancy alter the neighbor-based reasoning (e.g. biasing the underlying similarity) ?

**Additional Comments On Reviewer Discussion:**

The reviewers regret that the results tend to be over-claimed ("prove" is consistently used for "experimentally demonstrate" in the submitted version) and ambiguous (the definition of neighbor-based reasoning is not cristal clear).
The authors will tone down their claims.
They provided complementary results, thanks Rev ctTK, showing the impact of fine-tuning on inductive performances. It remains to also assess the impact of partial fine-tuning (e.g. fine-tuning the model on a single type of synthetic data, such as list transformation, and then evaluating its performance on other scenarios).

---

### Decision · Program_Chairs · 2025-01-22

Accept (Poster)